# MCL-1 safeguards activated hair follicle stem cells to enable adult hair regeneration

Hui San Chin [1] ✉, Jinming Cheng [2,3,4], Shih Han Hsu[1], Guo Guang Lum[1], Maria TK Zaldivia[3,4], Sarmilla Nelameham[1], Fusheng Guo[1], Keerthana Mallavarapu[1], Felicity C. Jackling [3], Jicheng Yang [3,4], Jonathan S. L. Tan[5], Prabha Sampath[1,5,6], Nick Barker [7,8], Gordon K. Smyth [2,9], Geoffrey J. Lindeman [3,4,10,11], Andreas Strasser [4,12], Jane E. Visvader [3,4], Yunshun Chen [2,3,4], Ting Chen [13] & Nai Yang Fu [1,3,4,8] ✉

Hair follicles cycle through expansion, regression and quiescence. To investigate the role of MCL-1, a BCL-2 family protein with anti-apoptotic and apoptosis-unrelated functions, we delete *Mcl-1* within the skin epithelium using constitutive and inducible systems. Constitutive *Mcl-1* deletion does not impair hair follicle organogenesis but leads to gradual hair loss and elimination of hair follicle stem cells. Acute *Mcl-1* deletion rapidly depletes activated hair follicle stem cells and completely blocks depilation-induced hair regeneration in adult mice, while quiescent hair follicle stem cells remain unaffected. Single-cell RNA-seq profiling reveals the engagement of P53 and DNA mismatch repair signaling in hair follicle stem cells upon depilation-induced activation. *Trp53* deletion rescues hair regeneration defects caused by acute *Mcl-1* deletion, highlighting a critical interplay between P53 and MCL-1 in balancing proliferation and death. The ERBB pathway plays a central role in sustaining the survival of adult activated hair follicle stem cells by promoting MCL-1 protein expression. Remarkably, the loss of a single *Bak* allele, a pro-apoptotic *Bcl-2* effector gene, rescues *Mcl-1* deletion-induced defects in both hair follicles and mammary glands. These findings demonstrate the pivotal role of MCL-1 in inhibiting proliferation stress-induced apoptosis when quiescent stem cells activate to fuel tissue regeneration.

The mammalian hair follicle (HF) serves as a valuable model for studying tissue patterning, regeneration, as well as stem cell biology. In mice, HF morphogenesis begins with placode emergence during fetal skin development (around E14.5), reaching completion approximately 6 days after birth[1,2]. Mature HFs are divided into two main compartments: the epithelial (epidermal) and dermal compartments. The epithelial layer consists of interfollicular epidermis (IFE), infundibulum, isthmus, HF bulge, hair germ and hair shaft, while the dermal compartment consists of the dermal papilla (DP) and the dermal sheath[3]. Throughout life, HFs undergo a continuous cycle involving distinct phases: growth (anagen), regression (catagen), and quiescence (telogen). This dynamic process is driven by HF stem cells (HFSCs) located in the bulge niche[4,5]. The first postnatal hair cycle is relatively synchronized and occurs between p20 (first telogen stage) to p55 (second telogen stage) in mice. Subsequently, HF cycling only occurs in small patches. During anagen, HFSCs are activated and proliferate, forming the outer root sheath (ORS) that extends downward to fuel hair growth[6–8]. The ORS envelops the DP and transit-amplifying progenitor cells (TACs) in the hair matrix, which cycle rapidly before differentiating to form the inner root sheath (IRS) and hair shaft[9–12]. During

catagen, TACs within HF matrix undergo cell death, while the remaining epithelial strand regresses upward[10]. Long-lived HFSCs and slow-cycling ORS cells are spared from cell death. As HF transition from catagen to telogen, some HFSCs migrate toward the DP to form the hair germ (HG). In telogen, HFSCs remain quiescent, a state that can persist for months.

Apoptosis, a highly orchestrated form of programmed cell death, is crucial for normal tissue development, homeostasis, and regeneration throughout life. It also plays a critical role in eliminating pathogen-infected cells and those undergoing early neoplastic transformation. In HFs, apoptotic events during catagen have been visualized through intravital microscopy in live mice[13]. The genetic deletion of Sep4/ARTS, a modulator of apoptosis, in mice led to an expansion of the HFSCs compartment, improved wound healing and hair regeneration[14]. Interestingly, a recent study suggests that the interference of the HFSC niche, either by hair shaft loss or miniaturization, triggers calcium-dependent apoptosis[15]. However, the molecular mechanisms by which apoptosis governs different stages of HF development and hair regeneration remains poorly understood[16,17].

The apoptotic pathway is tightly regulated by the BCL-2 family of proteins, which controls the mitochondrial outer membrane permeability and integrity, ultimately governing cell death decisions[18,19]. This family consists of pro-survival proteins (BCL-2, MCL-1, BCL-XL, BCL-W, BFL-1/A1), pro-apoptotic BH3-only proteins (BIM, PUMA, BID, BAD, NOXA, HRK, BIK, BMF), and multi-domain apoptotic effectors (BAX, BAK, BOK)[20]. When apoptosis is triggered, BH3-only proteins are upregulated, neutralizing pro-survival proteins and activating BAX and BAK[21]. These activated proteins oligomerize, causing mitochondrial outer membrane permeabilization (MOMP), which releases apoptogenic factors and initiates caspase activation (including executioner caspases−3, −6 and −7), culminating in a well-coordinated cell demolition.

While genetic models have elucidated the diverse roles of individual BCL-2 family members in various tissues[22], none has identified an essential physiological role in HF development or HFSC function. Although pharmacological inhibition of BCL-2 depleted supra-basal bulge stem cells, genetic deletion of Bcl-2 did not disrupt HF growth[23]. Nonetheless, deletion of Bcl-2 sensitized HFSCs to DNA-damage induced apoptosis and caused premature hair graying in mice, due to melanocyte depletion[24–27]. No skin or hair related defects have been reported in A1/Bfl-1[28] or Bcl-w knockout mice[29,30]. Among the pro-survival molecules, MCL-1 and BCL-XL are critical for embryonic development, and their respective knockouts cause embryonic lethality[31,32]. Given that Mcl-1 and Bcl-x knockout causes embryonic lethality respectively, conditional knockout systems are used to investigate the roles of MCL-1 and BCL-XL in various tissues. Although Keratin 5 (K5)-Cre mediated Bcl-x deletion did not affect hair regeneration, it increased keratinocyte sensitivity to UV-induced apoptosis[33]. Despite insights from knockout models, the molecular mechanisms governing HFSC survival, HF cycling, and hair regeneration remain unclear. Studies from conditional knockout mouse models have revealed that MCL-1 is essential for the survival of numerous cell types (as reviewed in[34]). Interestingly, MCL-1 has also been attributed several apoptosis-unrelated roles including regulating mitochondrial morphology and cell cycle, promoting pluripotency of stem cells, and accelerating long-chain fatty acid oxidation[34–39]. In this study, we unveiled the crucial role of MCL-1 in HFSC regulation and hair regeneration in mice.

## Results

### Dynamic expression of BCL-2 family proteins throughout the adult HF cycle

To elucidate the role of BCL-2 family proteins in HF development and hair regeneration, we analyzed the expression patterns of MCL-1, BCL-XL, BIM and BAK in mouse HFs at various stages of the HF cycle. The pro-survival proteins MCL-1 and BCL-XL were expressed during telogen and anagen but downregulated in the apoptotic strand during catagen (Figs. 1A, B, S1A, B). Interestingly, MCL-1 is ubiquitously expressed throughout the HF epithelium, while BCL-XL is mainly expressed in the IRS. Notably, the hair matrix, composed of progenitor TACs marked by PCAD, expresses high levels of MCL-1 but not BCL-XL during anagen, suggesting that MCL-1 may play a role in maintaining these highly proliferative cells during active hair growth. Expression of both MCL-1 and BCL-XL expression is restricted to the HF epidermal cells and absent in the DP throughout the adult HF cycle.

Conversely, the pro-apoptotic proteins BIM and BAK were readily detected in HFs during telogen and anagen (Figs. 1C, D, S1C, D). Their expression remains within the apoptotic strand during catagen. Unlike the pro-survival proteins MCL-1 and BCL-XL, pro-apoptotic BIM and BAK are also expressed in the DP. These results highlight the dynamic regulation and expression of distinct pro-survival and pro-apoptotic BCL-2 family members throughout the HF cycle.

### Deletion of *Mcl-1* in the skin epithelium leads to gradual hair loss in adult mice

To explore the role of MCL-1 in mouse skin and HFs, we employed a transgenic line expressing *Cre* recombinase under the control of a 5 Kb human *Krt5* (*K5*) promoter to delete *Mcl-1* in all skin epithelial lineages, including HFs (Fig. S1E, F). Using the *Rosa26-mTmG* reporter model, we confirmed that this *K5-Cre* system efficiently induced DNA recombination in the skin epithelium, but not in the DP (Fig. S1F, G, L), consistent with a previous report[40]. By crossing floxed *Mcl-1* mice with *K5-Cre* transgenic mice, we monitored the hair phenotype of offspring from birth (Fig. S1E, F). *K5-Cre/Mcl-1^{f/f}* mice lacking both *Mcl-1* alleles in the skin epithelium appeared runty after birth but ~80% of them survived into adulthood. Their hair growth was macroscopically normal during early HF organogenesis (p1-p12) (Fig. S2A, B) and up to the first anagen (p29), comparable to control (*Mcl-1^{f/f}* or *Mcl-1^{f/+}*) and *K5-Cre/Mcl-1^{f/+}* littermates (Fig. 2A, B). Histological analyses showed that HF morphogenesis was overall not affected while hair bulbs in *K5-Cre/Mcl-1^{f/f}* mice appeared smaller compared to either wildtype or *K5-Cre/Mcl-1^{f/+}* littermates (Figs. 2B, S2A). Additionally, distinct HF differentiation lineages marked by specific markers, including KRT14, LEF-1, GATA3, KRT6, and PCAD were comparable between *K5-Cre/Mcl-1^{f/f}* mice and control mice at p29 (Fig. S2C). These results indicate that MCL-1 is dispensable for HF morphogenesis during early development.

Interestingly, *K5-Cre/Mcl-1^{f/f}* mice showed gradual hair loss by around p60, eventually leading to complete hair loss by 5 months (Figs. 2A, B, S2D). In contrast, mice with a single *Mcl-1* allele deletion (*K5-Cre/Mcl-1^{f/+}*) did not show detectable HF defects. Histological sections indicated that hair loss in *K5-Cre/Mcl-1^{f/f}* mice was associated with HF miniaturization at p60, followed by the complete destruction of HFs by p90 (Fig. 2B). In our floxed *Mcl-1* mouse model, a truncated and non-functional form of hCD4 is incorporated into the mutant allele as a reporter of successful CRE-mediated recombination (*Mcl-1* deletion) and as a surrogate of endogenous *Mcl-1* promoter activity (Fig. S1E). Confocal imaging analyses revealed that hCD4 is universally expressed in *K5-Cre/Mcl-1^{f/+}* skin epithelium (Fig. S1M). Quantitative analysis of cell-surface hCD4 expression by FACS analysis confirmed efficient recombination of the floxed *Mcl-1* allele in HFSCs (Lin⁻/CD49f⁺/CD34⁺/SCA-1⁻), IFE (Lin⁻/CD49f⁺/SCA-1⁺) and the non-defined HF keratinocyte population (Lin⁻/CD49f⁺/CD34⁻/SCA-1⁻) (Figs. S2E, 2C) in *K5-Cre/Mcl-1^{f/f}* and *K5-Cre/Mcl-1^{f/+}* mice. These data also indicate that the *Mcl-1* promoter is active in all skin epithelial lineages, with the highest activity in the IFE layer, consistent with data from qPCR analysis (Fig. 2D). The deletion efficiency of the floxed *Mcl-1* allele in our conditional knockout model was further validated by qPCR (Figs. 2D, S2F). Confocal imaging analysis also showed a complete absence of MCL-1 in the HF

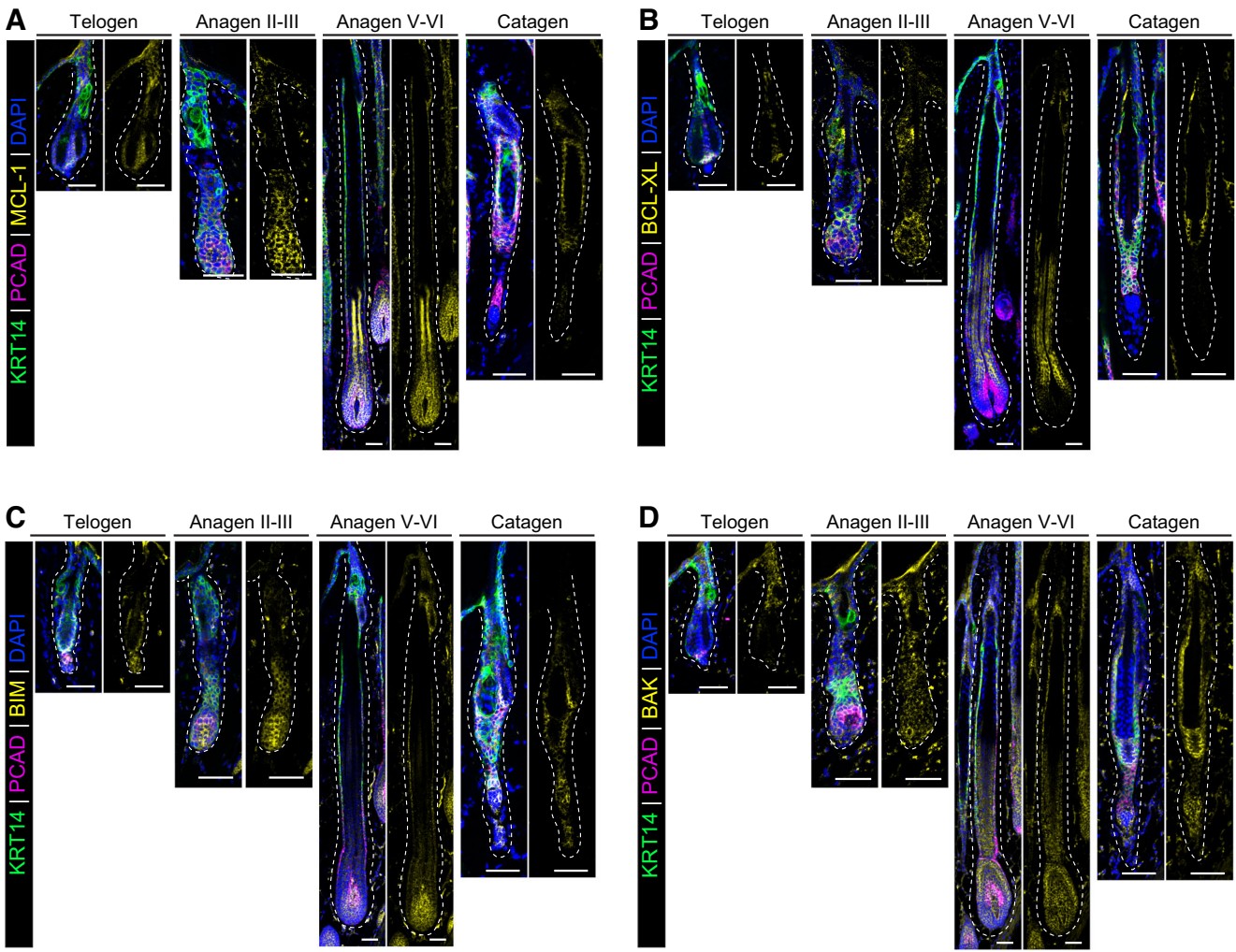

**Fig. 1 | Dynamic expression of BCL-2 family members at different stages of the HF cycle.** Representative confocal images showing the expression of MCL-1 (**A**), BCL-XL (**B**), BIM (**C**) and BAK (**D**) at p60 (telogen), p22 (anagen II-III), p29 (anagen V-VI) and p40 (catagen). Basal skin layer is stained by KRT14, hair matrix containing TACs is stained by PCAD and DAPI is used as a nuclear counterstain in each image. Data shown are representative of $n = 3$ independent experiments. Scale bars, 50 μm.

epithelium of *K5-Cre/Mcl-1^{f/f}* mice as compared with either *Mcl-1^{f/f}*, *Mcl-1^{f/+}* or *K5-Cre/Mcl-1^{f/+}* control littermates (Figs. 2E, S2A). Importantly, *Mcl-1* deficiency in the skin epithelium exhibited augmented levels of apoptotic cells, with numerous cells positive for activated (i.e. cleaved) CASPASE-3 (CC3) in HFs (Fig. 2F, G). Once again, the presence of a single allele of *Mcl-1* was sufficient to prevent apoptosis (Fig. S2G). These results highlight an obligate role of MCL-1 for the maintenance of adult HFs.

### MCL-1 is essential for the survival of adult CD34⁺ HFSCs
CD34⁺ cells within murine hair bulge are recognized as the putative quiescent HFSCs with multipotent potential that drive HF regeneration and hair growth. FACS analysis revealed a significant loss of CD34⁺ HFSCs (Lin⁻/CD49f⁺/CD34⁺/SCA-1⁻) in *K5-Cre/Mcl-1^{f/f}* mice at p90 compared to littermate controls (Fig. 3A, B). This reduction was also observed as early as p60 but not p21, consistent with histological sections showing HF defects in *K5-Cre/Mcl-1^{f/f}* mice at these ages; no such defects were seen in any of the control littermates (Fig. 2B). Notably, the presence of a single *Mcl-1* allele prevented the loss of the CD34⁺ HFSCs (Fig. 3B). Similarly, the non-defined HF keratinocyte population (Lin⁻/CD49f⁺/CD34⁻/SCA-1⁻) was significantly diminished in *K5-Cre/Mcl-1^{f/f}* mice at p90, while the IFE keratinocyte subpopulation (Lin⁻/CD49f⁺/CD34⁻/SCA-1⁺) increased significantly, correlating with complete HF loss

(Fig. 3A, B). This is in line with the histological analyses of skins from p90 *K5-Cre/Mcl-1^{f/f}* mice showing complete loss of the HF structures (Fig. 2B).

Western blot analysis of FACS-sorted HF and IFE cells at telogen revealed higher expression of pro-survival BCL-2 family proteins in HF and IFE (Lin⁻/CD49f⁺/SCA-1⁻ and Lin⁻/CD49f⁺/SCA-1⁺ respectively) compared to non-HF epidermal cells (Lin⁻/CD49f⁻) (Fig. 3C). This is in line with the FACS analysis of hCD4 expression as an indicator *Mcl-1* promoter activity and qPCR analysis of *Mcl-1* mRNA (Fig. 2C, D). Confocal imaging analyses confirmed the expression of MCL-1 and BIM in the HFSCs marked as CD34⁺ and KRT15 (K15⁺) (Fig. 3D). While the establishment of quiescent CD34⁺ HFSCs was not affected in young *K5-Cre/Mcl-1^{f/f}* mice at p21(Fig. 3A, B), we found that the ex vivo clonogenic capacity of these cells was significantly reduced compared to controls (Fig. 3E). This demonstrates that MCL-1 is essential for HFSCs to proliferate and form colonies under the in vitro culture condition tested.

### Acute deletion of *Mcl-1* in adult skin epithelium abrogates hair regeneration
In the *K5-Cre/Mcl-1^{f/f}* model, *Mcl-1* is constitutively deleted from embryonic stages of HF morphogenesis. To investigate MCL-1 function postnatally, we utilized the *K5-Cre^{ER}* line, allowing temporal tamoxifen-inducible *Mcl-1* deletion (Fig. S1F). *Mcl-1* deletion was induced at

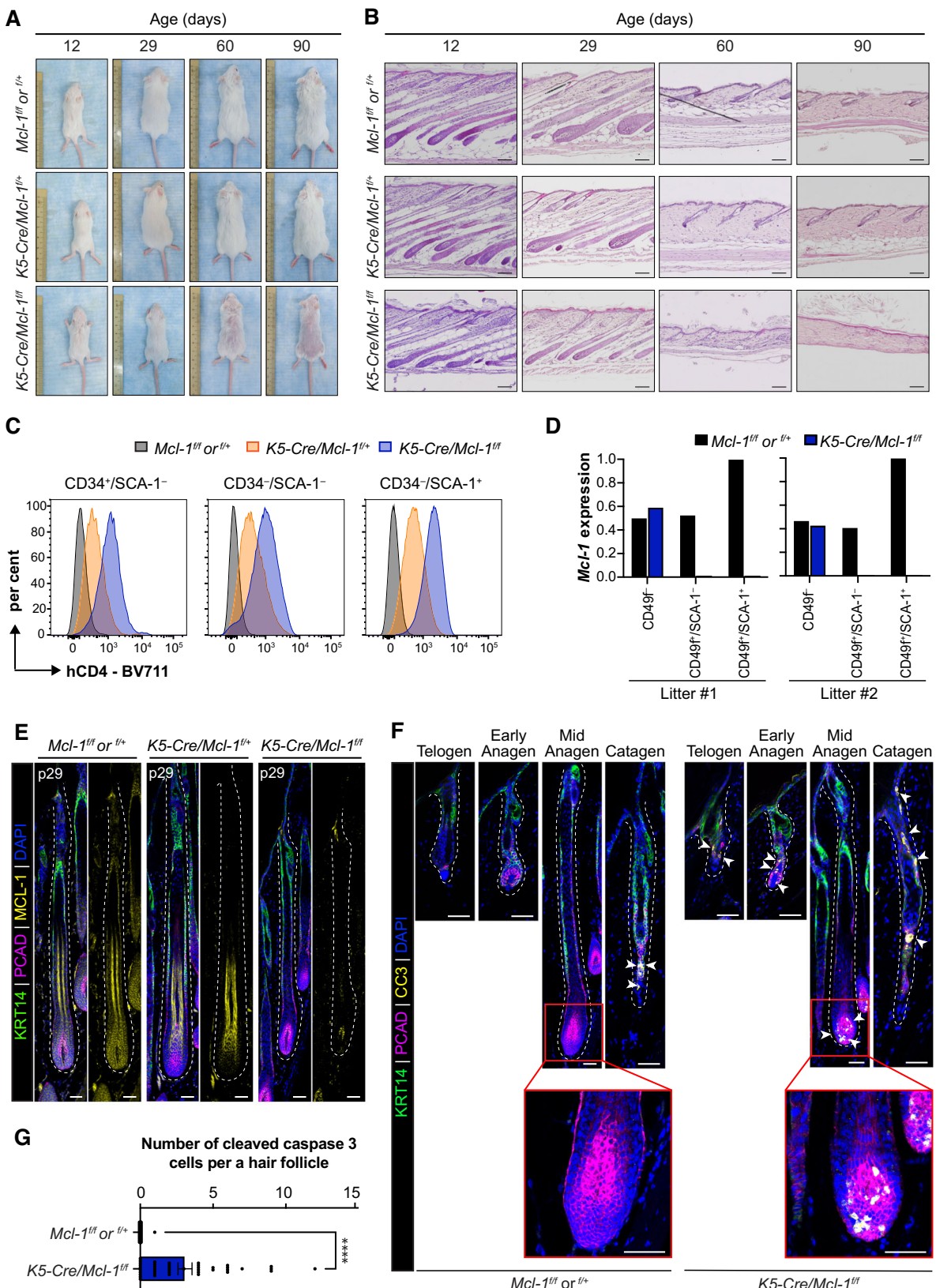

the second telogen (p55), and hair loss was monitored. Consistent with the constitutive *K5-Cre/Mcl-1^{f/f}* model, tamoxifen-induced *Mcl-1* deletion resulted in gradual hair loss, leading to the complete loss of HFs by 3 months (Fig. S3A). We then evaluated hair regeneration following depilation with *Mcl-1* deletion induced at the second telogen (Fig. 4A). While control mice fully regrew their hair within 4 weeks

post-depilation (wpd), *K5-Cre^{ER}/Mcl-1^{f/f}* mice exhibited no hair regrowth within the depilated area, and gradual hair loss extended to non-depilated regions (Fig. 4B). When *Mcl-1* was deleted later (p84), after murine HFs exit the synchronized hair cycle, hair regeneration defects were also observed (Fig. S3B). H&E staining confirmed HF regeneration defect by 9 days post depilation (dpd), with complete HF loss by

**Fig. 2 | MCL-1 is essential for murine HF during adulthood. A, B** Deletion of *Mcl-1* in the skin epithelium leads to progressive hair loss and HF destruction. Representative photographs of *Mcl-1^f/f^* or *Mcl-1^f/+^*, *K5-Cre/Mcl-1^f/+^* and *K5-Cre/Mcl-1^f/f^* mice at p12, p29, p60 and p90 (**A**), alongside H&E sections of dorsal skin at these time points (**B**). Data represent *n* = 8 mice per genotype. Scale bars, 100 μm.
**C** Representative FACS plots demonstrating efficient deletion of the floxed *Mcl-1* gene, using hCD4 as an indicator of efficient CRE-mediated recombination as well as promoter activity. Expression of hCD4 in hair bulge (CD49f⁺/CD34⁺/SCA-1⁻), non-bulge HF cells (CD49f⁺/ CD34⁻/SCA-1⁻) and interfollicular epidermis (IFE) (CD49f⁺/CD34⁻/SCA-1⁺) from *Mcl-1^f/f^* or *Mcl-1^f/+^*, *K5-Cre/Mcl-1^f/+^ and K5-Cre/Mcl-1^f/f^* mice. *n* = 3 at p21 per genotype. **D** qPCR analysis reveals a significant reduction in *Mcl-1* transcript levels in HF cells (CD49f⁺/SCA-1⁻) and IFE (CD49f⁺/SCA-1⁺), but not in non-epithelial cells (CD49f⁻) of *K5-Cre/Mcl-1^f/f^* mice at p21, compared to control

littermates. Data are presented as mean of three technical replicates for each point from two independent litters. **E** Representative confocal images show the absence of MCL-1 protein in HFs of *K5-Cre/Mcl-1^f/f^* mice compared to *K5-Cre/Mcl-1^f/+^* and *Mcl-1^f/f^* or *Mcl-1^f/+^* littermate controls at p29 (mid anagen). Data represents *n* = 5 experiments. Scale bar, 50 μm. **F** *Mcl-1* deletion leads to apoptosis in HFs, as shown by CC3 staining at various stages: telogen (p60), early anagen (p22), mid anagen (p29), and catagen (p40), representative of *n* = 5 independent mice per genotype. Basal skin is marked by KRT14, hair matrix with TACs is stained with PCAD, and DAPI is used as nuclear counterstain. Scale bar, 50 μm. **G** Quantification of CC3⁺ cells per HF at p29 shows significantly increased apoptosis in *K5-Cre/Mcl-1^f/+^* mice (*n* = 70) vs. controls (*n* = 40). Data are presented as mean ± SEM for *n* = 3 mice per genotype. ****$p < 0.0001$, Student's *t*-test.

11 weeks (Figs. 4C, S3C), highlighting the critical role of MCL-1 in initiating hair regrowth in adult mice.

To explore the mechanisms underlying these defects, we performed immunofluorescent staining for CC3 and PCNA. *Mcl-1* deletion led to marked apoptosis (CC3⁺) at 3 and 9 dpd, with reduced PCAD⁺ cell population at 3, 6, and 9 dpd (Figs. 4D, S3C, S4A). PCNA staining indicated no significant changes in proliferation (Figs. 4E, S5E), suggesting that apoptosis, rather than a proliferation defect, is responsible for the impaired HF regeneration. Importantly, *Mcl-1* deletion caused a marked reduction in CD34⁺ HFSCs as early as 3 dpd, with further depletion at 9 dpd (Figs. 4F, S4B). Other HFSC markers, including KRT15[41], SOX9[42], and LHX2[43], were also significantly reduced in *Mcl-1*-deficient HFs compared to controls (Fig. S4C–E), while LRIG1[44], a marker of IFE stem cells, remained comparable (Fig. S4F). Flow cytometry confirmed a significant reduction in CD34⁺ HFSCs by 6 dpd, and these cells were nearly absent by 11 wpd (Fig. 4G–J). We next examined whether MCL-1 is necessary for the short-term survival of quiescent HFs. Tamoxifen-induced *Mcl-1* deletion at around p55 without depilation revealed no significant impact on quiescent HFs including HFSCs, as shown by histological confocal analyses and flow cytometry analysis (Fig. S3D–G). These findings demonstrate that MCL-1 is an essential pro-survival protein for activated HFSCs during early hair regeneration in adult mice.

As MCL-1 is also expressed in the IFE (Figs. 1A, 2C, D, 3C), we assessed whether acute deletion of *Mcl-1* affected skin stratification, barrier function, and overall integrity. TransEpidermal Water Loss (TEWL) measurement showed no significant differences post depilation, and markers of stratification (CLAUDIN-1 and KRT10) and barrier integrity (PLECTIN, DESMOPLAKIN, E-CADHERIN, LORICRIN, and FILLAGRIN) were all unchanged between *K5-Cre^ER^/Mcl-1^f/f^* and control littermates (Fig. S5A–D), suggesting that other pro-survival BCL-2 family proteins likely compensate for the loss of *Mcl-1* in the IFE. Additionally, inflammation was not observed, with comparable levels of CD4⁺ and CD8⁺ T cells, and F4/80⁺ macrophages across genotypes (Figs. 4C, S5F).

HFSCs rapidly lose their basal markers (e.g., K5 and K14) upon activation and differentiation into TACs to drive HF regeneration[45]. We explored whether *Mcl-1* deletion in other skin epithelial cells expressing basal cell markers, including IFE cells, results in HF defects after HFSCs have been activated and committed to TACs. *Mcl-1* deletion was induced after HFSC activation in *K5-Cre^ER^/Mcl-1^f/f^* mice by administrating tamoxifen-containing diet at 3 dpd. Hair regeneration was not impaired under this condition (Fig. S4G), indicating that the loss of *Mcl-1* disrupts HF regeneration only when it is deleted before HFSCs commit to more differentiated lineages. Confocal imaging confirmed efficient deletion of *Mcl-1* in the IFE and ORS of HF (Fig. S4H–J) while the integrity and structure of IFE were maintained, and CC3 activation was not detected (Fig. S4K).

The HFSC compartment is heterogenous and *Lgr5* expression has been shown to mark a subset of HFSCs[46]. We generated an

tamoxifen-inducible knockout model, *Lgr5-Cre^ER^/Mcl-1^f/f^*/tdTomato (Fig. S1F, H), to specifically assess the role of MCL-1 in the *Lgr5*⁺ HFSC subpopulation during adult hair regeneration[47]. To achieve maximum deletion of *Mcl-1* in this model, mice were placed on tamoxifen-containing diet from −2 days to 15 dpd (Fig. S6A). Under this experimental condition, approximately 30% of CD34⁺ cells and -5% of non-defined HF cells expressed tdTomato and hCD4 (a surrogate of *Mcl-1* deletion), with no recombination in SCA-1⁺ IFE cells (Fig. S6B). Histological analysis revealed no differences in hair regeneration between *Lgr5-Cre^ER^/Mcl1^f/f^* and *Mcl-1^f/f^* controls (Fig. S6C). Confocal imaging analysis indicates that RFP⁺ cells are mainly localized at the lower bulge and proliferative ORS cells, but not at the matrix which contains TACs. Additionally, deletion of both *Mcl-1* alleles did not affect the distribution patterns and number of RFP⁺ cells in HFs (Fig. S6D). Cleaved caspase-3 staining was not detected when *Mcl-1* was deleted in *Lgr5*-expressing cells (Fig. S6D). Consistent with the FACS data, only a subset of CD34⁺ expressed RFP, suggesting that *Mcl-1* deletion occurred in a limited fraction of the HFSC population (Fig. S6E). The deletion of *Mcl-1* in the cells of bulge and ORS was further confirmed by MCL-1 and hCD4 staining (Fig. S6F, G). Collectively, these results indicate that MCL-1 is dispensable for the function and cell fate of *Lgr5*⁺ HFSCs during adult hair regeneration induced by depilation.

## Analysis of the adult HF regeneration dynamics by single-cell RNA-seq

As the acute deletion of *Mcl-1* dramatically reduced the CD34⁺ HFSC population only upon HF activation, we focused on the molecular and cellular dynamics of adult HF regeneration induced by hair depilation. To explore the full repertoire of cell types and their gene expression profiles at various stages of adult HF regeneration, we isolated cells from the dorsal skin of two mice at different time points (0 h, 4 h, 6 h, 2 d, 4 d and 6 d post depilation) and performed single-RNA (scRNA)-seq profiling. Transcriptomes of 48,777 valid single cells in six dorsal skin samples were integrated and clustered and then cell types were annotated based on known markers from previous studies[48,49]. The analysis identified 11 cell clusters including: 1. dermal papilla (DP) and fibroblast; 2. upper HF basal cells (uHF.B); 3. upper HF suprabasal cells (uHF.SB); 4. T cells; 5. Langerhans; 6. Melanocytes; 7. sebaceous glands (SG); 8. outer bulge cells (OB); 9. interfollicular epidermal basal cells (IFE.B); 10. interfollicular epidermal suprabasal cells (IFE.SB); 11. Other−a small subset yet to be defined (Fig. S7A, B). The OB and uHF.B cells could be further divided into cycling and non-cycling subpopulations. The cycling_uHF.B population was present across the six different time points. However, the number of cycling_OB cells markedly increased from 2 dpd, peaking at 4 dpd (Fig. S7C, D). Analysis of differentially expressed (DE) genes in each cluster across the six time points of HF regeneration revealed approximately 800 and 500 DE genes in OB and melanocytes, respectively, while only a limited number of DE genes were identified in the remaining cell clusters (Fig. S7E).

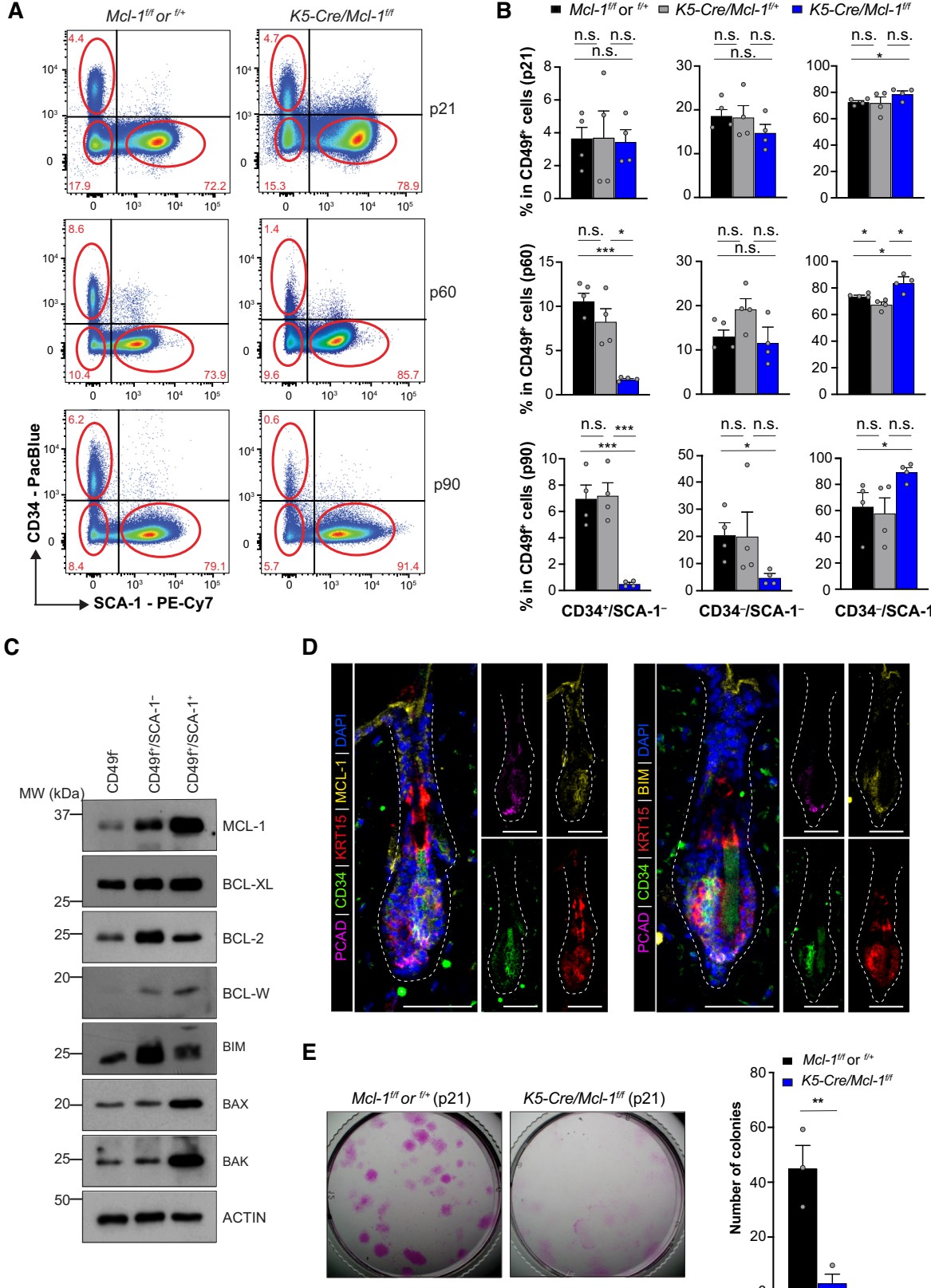

### Enrichment of the P53 and DNA mismatch repair signaling in activated HFSCs

To further define HF cell lineages, we focused on HF cells across different time points and re-clustered them (Fig. 5A, B). This analysis identified eleven distinct clusters within the 18,956 HF cells, each expressing specific markers and defined by unique gene expression signatures (Figs. 5C, S8A). Notably, OB cells were further sub-clustered into three subsets (OB1/OB2/OB3; Fig. 5A–C), and three IRS clusters emerged at 4 dpd (Fig. 5D). Interestingly, the OB1 subset, comprising both cycling and non-cycling OB1 cells, showed the highest number of differentially expressed (DE) genes during the time course of HF regeneration (Fig. 5D, E). A heat map of the top 100 DE genes revealed

**Fig. 3 | The maintenance of adult CD34+ HF bulge stem cells is dependent on MCL-1. A, B** Deletion of *Mcl-1* resulted in the gradual reduction of the CD34+ HF bulge cell subset. Representative FACS plots for skin epithelial cells fractionated based on CD34 and SCA-1 expression from *K5-Cre/Mcl-1^(f/f)* mice or littermate control mice at p21, p60 and p90 (*n* = 4) (**A**). Bar graphs showing the percentages of the indicated cell sub-populations within the CD49f+ epithelium isolated from the dorsal skin of *K5-Cre/Mcl-1^(f/f)*, *K5-Cre/Mcl-1^(f/+)*, *Mcl-1^(f/f)* or *Mcl-1^(f/+)* mice at p21, p60 and p90. Data are presented as mean ± SEM (*n* = 4 mice for each genotype). *\*p < 0.05; \*\*p < 0.01; \*\*\*p < 0.001; n.s., non-significant; Student's *t*-test (**B**). **C** Western blot analysis of the indicated BCL-2 family members in the CD49f⁻ (non-epithelial cells), CD49f+/SCA-1⁻ (HFs), CD49f+/SCA-1+ IFE populations of HFs at telogen (p60). Data shown are representative of *n* = 4 independent experiments. **D** Representative confocal images showing the expression of MCL-1 and BIM protein in HF stem cells marked by CD34+ and KRT15+ in hair bulges at telogen phase (p60). Representative of *n* = 3 experiments. Scale bar, 50 µm. Hair matrix containing TACs is stained by PCAD, and DAPI is used as a nuclear counterstain in each image. **E** In vitro clonogenic assay of the CD34+ HF bulge stem cells. 100 Lin⁻/CD49f+/CD34+/SCA-1⁻ cell-sisolated from the skin of *K5-Cre/Mcl-1^(f/f)* mice and littermate control mice at p21 were grown on 3T3 feeder cells for 14 days. Bar graphs show the numbers of colony-forming cells among the seeded CD34+ HF bulge cells. Data are presented as mean ± SEM for *n* = 3 independent experiments. *\*\*p < 0.01; Student's *t*-test.

significant transcriptional changes from 2 dpd onwards (Fig. S8B), with rapid upregulation of genes such as *Cgref1*, *Ets1*, *Rin1*, *Krt90*, and *Krt8* post depilation, followed by a return to baseline.

Interestingly, *Fgf18*, known to be downregulated during HF regeneration[50–52], was expressed in both OB1 and OB2 subsets, with a specific decrease in OB1 cells. The Hedgehog pathway, which includes both *Gli1* and *Shh*, plays a critical role in initiating HF regeneration[45,53]. In our analysis, the number of *Gli1*+ cells within the OB1 subset increased, while *Shh* expression was exclusively activated in the IRS2 subset starting at 2 dpd (Fig. S8C). KEGG enrichment analysis of the ~800 DE genes in OB1 cells across six time points revealed upregulated genes related to the cell cycle, DNA replication, repair, senescence, and the P53 signaling pathway (Fig. S8D, left). In contrast, downregulated genes were associated with focal adhesion, cell junctions, ECM-receptor interactions, and pathways regulating stem cell pluripotency (Fig. S8D, right). Additionally, P53 signaling and DNA mismatch repair pathways were specifically upregulated in cycling OB1 cells compared to non-cycling OB cells (Fig. 5F). To explore lineage relationships, we used RNA velocity analysis, which suggested that uHF.B cells may transition to either Cycling uHF.B or uHF.SB cells, indicating uHF.B cells as a possible origin (Fig. S8E). Moreover, a subset of cycling OB cells located at the boundary between the OB1 and Cycling_OB1 clusters showed transitional properties, potentially representing activated HFSCs re-entering the cell cycle.

Although hair germs (HGs) are known to play a key role in initiating adult HF regeneration, they were not clearly identified in the initial analysis, likely due to their low abundance across samples. By reintegrating data from the 0 h, 4 h, and 16 h timepoints, we identified a distinct HG cell cluster (Fig. S9A–D). Differential expression analysis revealed significant gene expression changes as early as 4 h post depilation. A combined analysis of the 4 h and 16 h samples versus 0 h identified 693 upregulated and 578 down-regulated genes at FDR < 0.05 (Fig. S9E). KEGG enrichment analysis showed upregulation of genes related to the proteasome, spliceo-some, metabolic pathways, actin cytoskeleton regulation, and DNA replication, while TNF, TGF-beta, and MAPK signaling pathways were downregulated (Fig. S9F). Additionally, a new OB cluster, OB4, emerged in the 4 h and 16 h samples but not in quiescent HFs (Fig. S9D). Most OB4 cells were originally classified as OB1 in Fig. 5A, indicating substantial transcriptional changes in OB1 cells during early HF regeneration.

### *TrpS3* deficiency rescues the HF defects caused by acute *Mcl-1* deletion

The upregulation of cell cycle, DNA replication and mismatch repair, senescence, and P53 signaling pathways in cycling OB1 cells (Figs. 5F, S8D, Left) implies the activation of DNA damage response may occur in HFSCs upon depilation-induced activation. Confocal microscopy identified γH2AX (p-H2A.X) foci in HFs, particularly in PCAD+ hair germ regions at 2, 4, and 6 dpd, absent in unshaved control littermates (Fig. 6A). Importantly, a subset of CD34+ and KRT15+ HFSCs also exhibited pH2A.X staining, indicating the presence of DNA

damage within the HFSC population during the early stages of hair regeneration (Fig. 6B). This further demonstrates the proliferation stress responses during early hair regeneration.

To examine the role of P53 in proliferation stress of activated HFSCs, we generated *K5-Cre^(ER)/Mcl-1^(f/f)/Trp53^(f/f)* mice and performed HF regeneration assays (Figs. 6C, S1F, K). Tamoxifen-induced deletion of *Mcl-1* and *Trp53* at p55 was confirmed by qPCR (Fig. 6D). H&E staining indicated that *Mcl-1*-deficient HFs failed to regenerate hair in the presence of one or both alleles of *Trp53* (*K5-Cre^(ER)/Mcl-1^(f/f)/Trp53^(+/+ or f/+)*) at 9 dpd (Fig. 6E). Remarkably, *Trp53* deletion in both alleles (*K5-Cre^(ER)/Mcl-1^(f/f)/Trp53^(f/f)*) significantly rescued HF defects in the *Mcl-1* deficient skin. Hair regrowth in *K5-Cre^(ER)/Mcl-1^(f/f)/Trp53^(f/f)* mice was comparable to littermate controls by 15 dpd and maintained through 30 dpd (Fig. 6F). Notably, *Trp53* deletion alone had no impact on hair regeneration in *K5-Cre^(ER)/Trp53^(f/f) and K5-Cre^(ER)/Mcl-1^(f/+)/Trp53^(f/f)* mice. We confirmed efficient CRE-mediated recombination through the appearance of hCD4 reporter and absence of endogenous MCL-1 in *K5-Cre^(ER)/Mcl-1^(f/f)/Trp53^(f/f)* mice (Fig. 6G, H). These findings demonstrate that P53 signaling may activate cell death, but MCL-1 conferred survival cues to HFSCs under proliferation stress.

### Concomitant deletion of *Bim* partially rescues the hair regeneration defect caused by the absence of *Mcl-1*

None of the Bcl-2 family genes were significantly altered during HF regeneration in our scRNA-seq analysis (Fig. S7H). The BH3-only genes *Bik*, *Hrk*, and *Bmf* were absent in most HF cells, while *Bad* and *Bnip3L* were universally expressed across HF cell clusters. Notably, *Bcl2l1*, *Pmaip1*, and *Bbc3* showed preferential expression in certain HF clusters. The apoptosis effector genes *Bax*, *Bak*, and *Bok* were widely expressed across HF clusters. Of the five pro-survival genes, all except *Bcl2a1a* were expressed in HF cells, with *Mcl-1* being relatively highly expressed in most HF cell types.

The BH3-only protein BIM plays a critical role in initiating apoptosis across various tissues and cell lines[54–56]. MCL-1 physically interacts with BIM[57], and this interaction is proposed to block apoptosis. Our data show that BIM is expressed in distinct cell lineages within the mouse HF (Figs. 1C, S7H). To assess contribution of BIM to defects caused by *Mcl-1* deletion during hair regrowth, we crossed *K5-Cre^(ER)/Mcl-1^(f/f)* with *Bim^(−/−)* mice to generate a double knockout (Fig. S1I, 7A). While HF regeneration was completely inhibited in *Mcl-1* knockout mice, hair regrowth was observed in *K5-Cre^(ER)/Mcl-1^(f/f)* with *Bim^(−/−)* mice, though not to the level of *Mcl-1^(f/f)* or *Mcl-1^(f/+)* control littermates (Fig. 7B, C). This suggests that while BIM is a key initiator of apoptosis upstream of MCL-1, other BH3-only proteins, also expressed in HF cells (Fig. S7H), likely play overlapping roles in neutralizing MCL-1 to trigger apoptosis in HF epidermal cells.

### Deletion of a single *Bak* allele fully restores the stem cell pool and rescue the defects in *Mcl-1*-deficient HFs and mammary glands

Mouse genetic studies indicate that the pro-apoptotic effectors BAX and BAK together are essential for early development and mito-chondrial-dependent apoptosis, though their functions appear

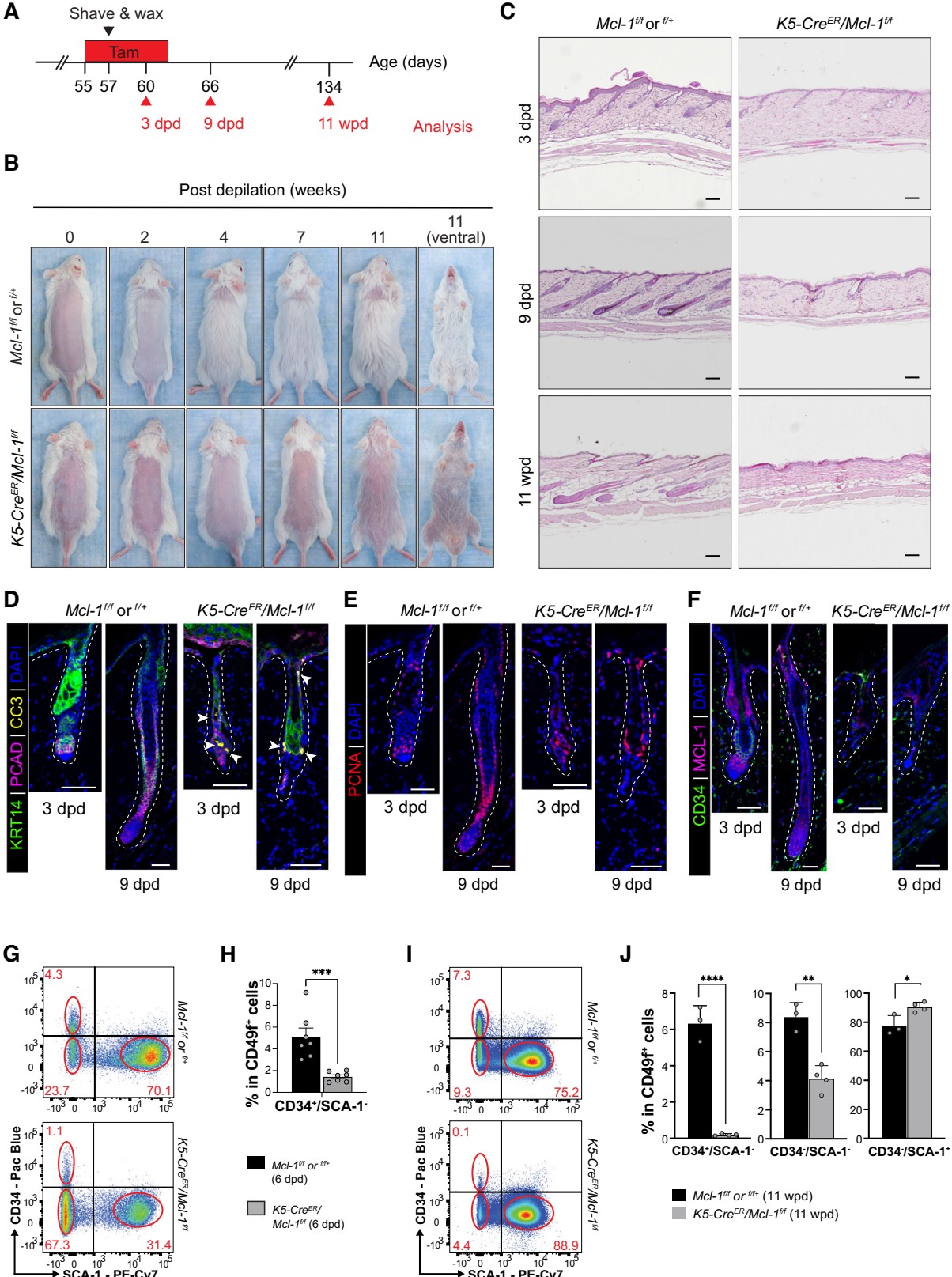

redundant in most cell types[58,59]. MCL-1 preferentially restrains BAK, but can also inhibit BAX, as shown primarily in vitro[60–62]. To investigate whether BAK is a downstream effector of MCL-1 in vivo, we generated *K5-Cre^ER/Mcl-1^f/f/Bak^−/−* mice (Figs. 7A, S1J). Remarkably, HFs from *K5-Cre^ER/Mcl-1^f/f/Bak^+/−* and *K5-Cre^ER/Mcl-1^f/f/Bak^−/−* mice regenerated comparably to littermate controls, indicating that loss of even a

single *Bak* allele prevented the hair regrowth defects caused by acute *Mcl-1* deletion (Fig. 7D, E). We also assessed whether *Bak* deletion rescues defects in constitutive *Mcl-1* knockout models. Loss of a single *Bak* allele fully prevented the hair loss and HF degeneration observed in *K5-Cre/Mcl-1^f/f* mice (Fig. 7F), restoring both CD34+ bulge HFSCs and non-defined HF cell populations, as confirmed by FACS

**Fig. 4 | Acute deletion of *Mcl-1* in the skin epithelium abrogates hair regeneration in adulthood. A** Experimental strategy for hair regeneration assay in adult *K5-Cre^ER/Mcl-1^f/f* mice. Deletion of *Mcl-1* was induced through administration of a tamoxifen containing diet between p55-p62. A portion of the dorsal skin was shaved and waxed to induce hair regeneration. Skin samples were harvested at the indicated timepoints and analyzed. **B** Representative images of mice taken at 0, 2, 4, 7, 11 wpd. *n* = 8–10 mice per genotype for each timepoint. **C** Representative H&E-stained histological sections of dorsal skin harvested at 3 dpd, 9 dpd and 11 wpd. *n* = 6–8 mice per genotype for each timepoint. Scale bars, 100 μm. **D–F** Confocal microscopy analysis of HFs from *K5-Cre^ER/Mcl-1^f/f* mice and littermate control mice at 3 dpd and 9 dpd. Representative confocal images depicting increased apoptosis indicated by CC3⁺ cells (**D**), proliferative cells as marked by PCNA staining (**E**), and the loss of CD34⁺ HFSC post-depilation (**F**) in *Mcl-1* deficient mice. *n* = 6–8 mice per genotype. Scale bars, 50 μm. Basal skin layer is stained by KRT14, hair matrix containing TACs is stained with PCAD and DAPI is used as a nuclear counterstain in each image in (**D–F**). **G–J** FACS analysis of dorsal skin epithelial cells at 6 dpd (**G, H**) and 11 wpd (**I, J**), fractionated by CD34 and SCA-1 expression. **H, J** Bar graphs show percentages of each cell sub-population within CD49f⁺ epithelium, mean ± SEM (*n* = 7 mice for 6 dpd; *n* = 3 for 11 wpd per genotype). ****$p < 0.0001$; **$p < 0.01$; *$p < 0.05$, Student's *t*-test.

(Fig. 7G, H). The absence of a single *Bak* allele was also able to restore the clonogenic potential of CD34⁺ HFSCs lacking *Mcl-1* (Fig. 7I, J). Collectively, these findings highlight the role of MCL-1 in restraining BAK-mediated apoptosis in HF survival and maintenance, HFSC population and hair regeneration.

Given that MCL-1 is essential for mammopoiesis and for repopulation by mammary stem cells (MaSC) upon transplantation[63,64], we investigated whether MCL-1 exerted a similar mechanism of action in this organ. In *Mcl-1* deficient glands, the quiescent Tspan8^high MaSC population was reduced by 85% (Fig. S10A–C). Concomitant deletion of a single *Bak* allele restored MaSC numbers and ductal morphogenesis (Fig. S10D, E), confirming a similar MCL-1/BAK regulatory mechanism in the mammary gland.

### Kinase screen identifies ERBB signaling as a regulator of MCL-1 protein expression in primary keratinocytes

To elucidate the regulatory mechanisms of MCL-1 expression in HFs, we performed a kinase inhibitor library screen on primary keratinocytes and assessed MCL-1 protein levels. MCL-1 expression was normalized and plotted, with pathways highlighted in blue when inhibition by kinase family inhibitors reduced MCL-1 levels below 0.5 (Fig. 8A). The screen identified several ERBB (EGFR) pathway inhibitors, including erlotinib, which significantly reduced MCL-1 protein levels in keratinocytes (Fig. 8B). Further validation with afatinib, a clinically-relevant ERBB inhibitor, confirmed that ERBB signaling regulates MCL-1 protein levels (Fig. 8C). Notably, afatinib treatment did not affect other BCL-2 family members, such as BCL-2, BCL-XL, BIM, BAK, or BAX, indicating a specific effect on MCL-1. qPCR analysis showed that afatinib did not alter *Mcl-1* mRNA levels, suggesting the downregulation occurs at the translational level (Fig. 8D).

Subsequent Western blot analysis revealed that afatinib reduced phosphorylation of p70 S6 kinase and its downstream targets, including S6, eIF4B, and eEF2K, in keratinocytes, indicating that ERBB signaling controls MCL-1 through protein synthesis pathways (Fig. 8E). To investigate the role of ERBB signaling in HFs, we conducted confocal microscopy imaging to determine the expression patterns of p-EGFR and p-ERBB2 in HFs throughout the HF cycle. However, the p-EGFR antibody did not work on the mouse skin under our experimental condition. p-ERBB2 expression in HFs was high during anagen, decreasing in catagen and remaining low in telogen, consistent with previous reports. However, p-ERBB2 expression was not localized to the K14⁺ outer root sheath or HFSC compartment, where MCL-1 is predominantly expressed during the anagen phase (Fig. S11A). This suggests that p-ERBB2 expression may not directly correlate with MCL-1 dynamics in HFSCs. Instead, the involvement of other ERBB-family members, such as EGFR, might play a more prominent role in regulating MCL-1 expression during HF regeneration. Interestingly, p-S6 expression, which marks translational activation downstream of ERBB signaling, followed a similar cycle-dependent pattern (Fig. S11B). These findings indicate that while ERBB signaling likely modulates MCL-1 protein levels through translational control in keratinocytes, the specific upstream ERBB receptor mediating this regulation warrants further investigation.

### Inhibition of the ERBB signaling pathway reduces MCL-1 protein expression and blocks HF regeneration

To evaluate the role of ERBB signaling in HF regeneration and its regulation of MCL-1 expression, we treated wild-type FVB/N mice with afatinib or vehicle control starting at p55, followed by dorsal skin depilation (Fig. 8F). Hair regeneration was notably inhibited in afatinib-treated mice compared to vehicle-treated controls at 9 dpd. Flow cytometry revealed a reduction in the CD34⁺ HFSC compartment (Lin⁻/CD49f⁺/CD34⁺/Sca-1⁻) in afatinib-treated mice, consistent with the phenotype observed in *Mcl-1*-deficient mice (Fig. 8G, H). Notably, afatinib did not reduce CD34⁺ HFSCs during quiescence (Fig. S11C). Afatinib treatment reduced MCL-1 protein levels (Fig. S11D), while *Mcl-1* mRNA levels in HF epidermal cells (Lin⁻/CD49f⁺/Sca-1⁻) were unaffected (Fig. S11E). Additionally, p-ERBB2 and p-S6 were significantly reduced in afatinib-treated mice (Fig. 8I, J). Increased apoptosis was evident by CC3 staining, though PCNA⁺ proliferating cells remained detectable (Fig. 8K). Treatment with erlotinib, another ERBB inhibitor in the kinase library, similarly blocked hair regeneration and decreased MCL-1 expression (Fig. S11F & G). These results suggest that ERBB signaling supports hair regeneration, at least in part, by promoting MCL-1 protein synthesis.

The ERBB family includes four receptors: EGFR/ERBB1, ERBB2, ERBB3, and ERBB4. While EGFR, ERBB3, and ERBB4 interact with various ligands, ERBB2 lacks direct ligand binding. Our scRNA-seq data show minimal ERBB4 expression in dorsal skin samples, with EGFR as the predominant ERBB receptor in OB cells (Fig. S7F, G). Neither ERBB receptors nor ligands were significantly modulated at the mRNA level across skin cell clusters during HF regeneration. These findings suggest that ERBB signaling remains constitutively active in OB cells, potentially activated by ligands from neighboring cell types, supporting HFSC function in both quiescent and activated states.

## Discussion

The essential roles of the BCL-2 protein family in the development and regeneration of various tissues and cell types have been extensively documented through mouse genetic models[22]. However, their specific functions in regulating HF morphogenesis, HF cycle, and HF regeneration during adulthood remain largely undefined. Previous studies using knockout mice revealed that none of the pro-survival proteins, including BCL-2, BCL-XL, BCL-W or BFL-1/A1, are essential for HF development and hair regeneration under physiological conditions. Here, we provide compelling evidence for the indispensable role of the pro-survival BCL-2 family member, MCL-1, in hair regrowth and the maintenance of HFSCs in adult mice.

Following HF organogenesis, long-term CD34⁺ quiescent HFSCs emerge during the first postnatal telogen stage (around p20) and are vital for subsequent HF cycles in adult mice. Our data show that the formation of these quiescent HFSCs at the end of HF organogenesis remains unaffected in the *Mcl-1*-deficient skin even though MCL-1 is universally expressed in HF epidermal cells during morphogenesis. However, *Mcl-1*-deficient mice undergo hair loss from adulthood, gradually progressing to complete hairlessness, accompanied by the elimination of HFs

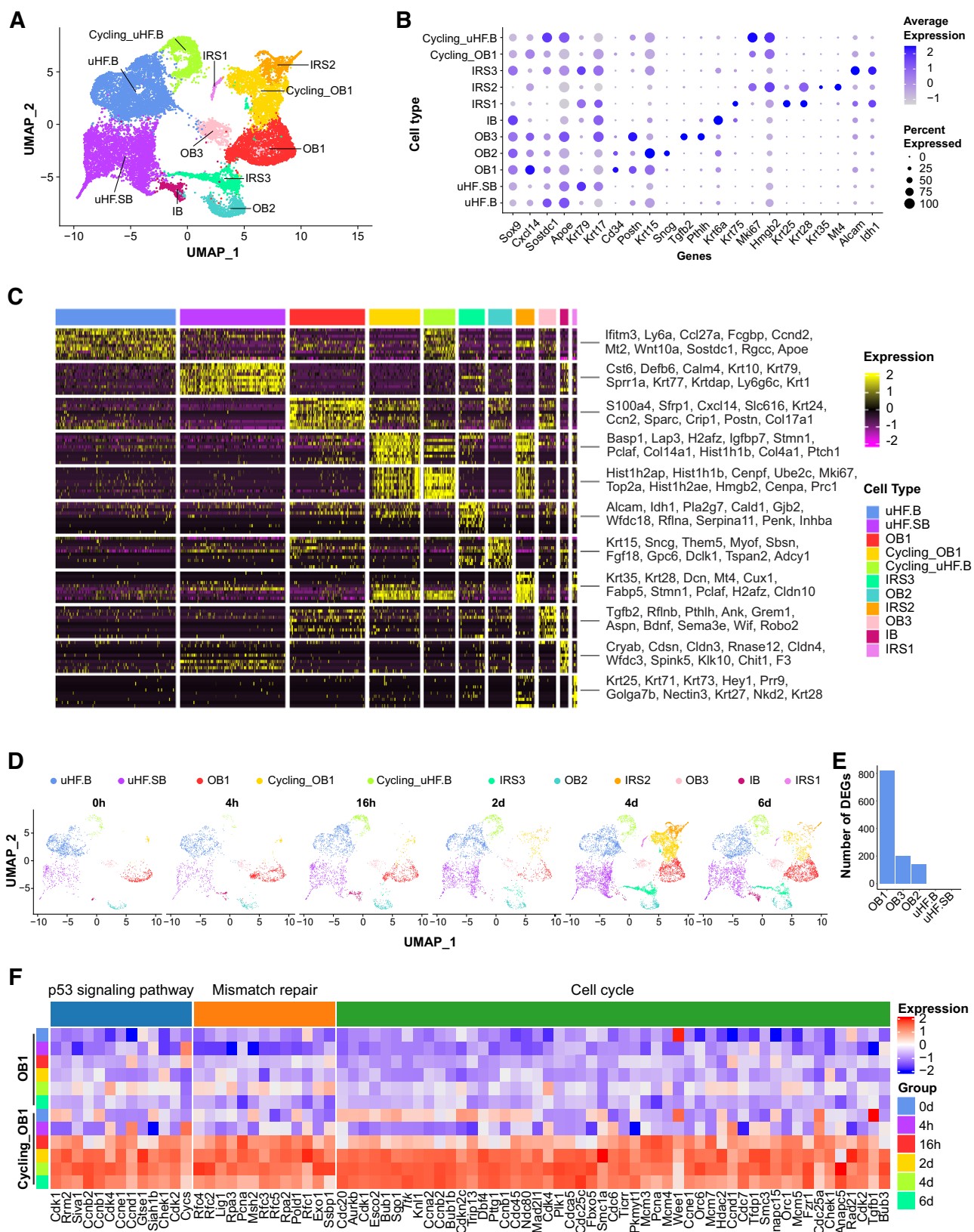

**Fig. 5 | Single-cell transcriptome atlas of HF cells from murine dorsal skin at early depilation-induced anagen.** **A** UMAP representation of all HF cells in the murine dorsal skin across six time points at the early anagen stages of hair regeneration after depilation, colored according to the main cell classes. The plot was generated by integration of 1732, 1627, 2342, 2500, 6322 and 4433 cells for 0 h, 4 h, 16 h, 2 d, 4 d and 6 d post-depilation, respectively. **B** Dot plot of marker genes used for annotation of cell clusters in (A). The colour scales correspond to the averaged expression levels. The dot size corresponds to the ratio of cells expressing the gene in the cell type. **C** Heatmap of top 10 marker genes for each cell cluster in (A). **D** UMAP visualization of HF cells in each time point post-depilation as indicated. **E** Number of differentially expressed (DE) genes for each cell type during hair regeneration along the time course. **F** A heatmap showing the upregulated and downregulated DE genes in the p53 signaling, mismatch repair and cell cycle pathways in cells from the OB1 and cycling_OB1 clusters over the course of hair regeneration.

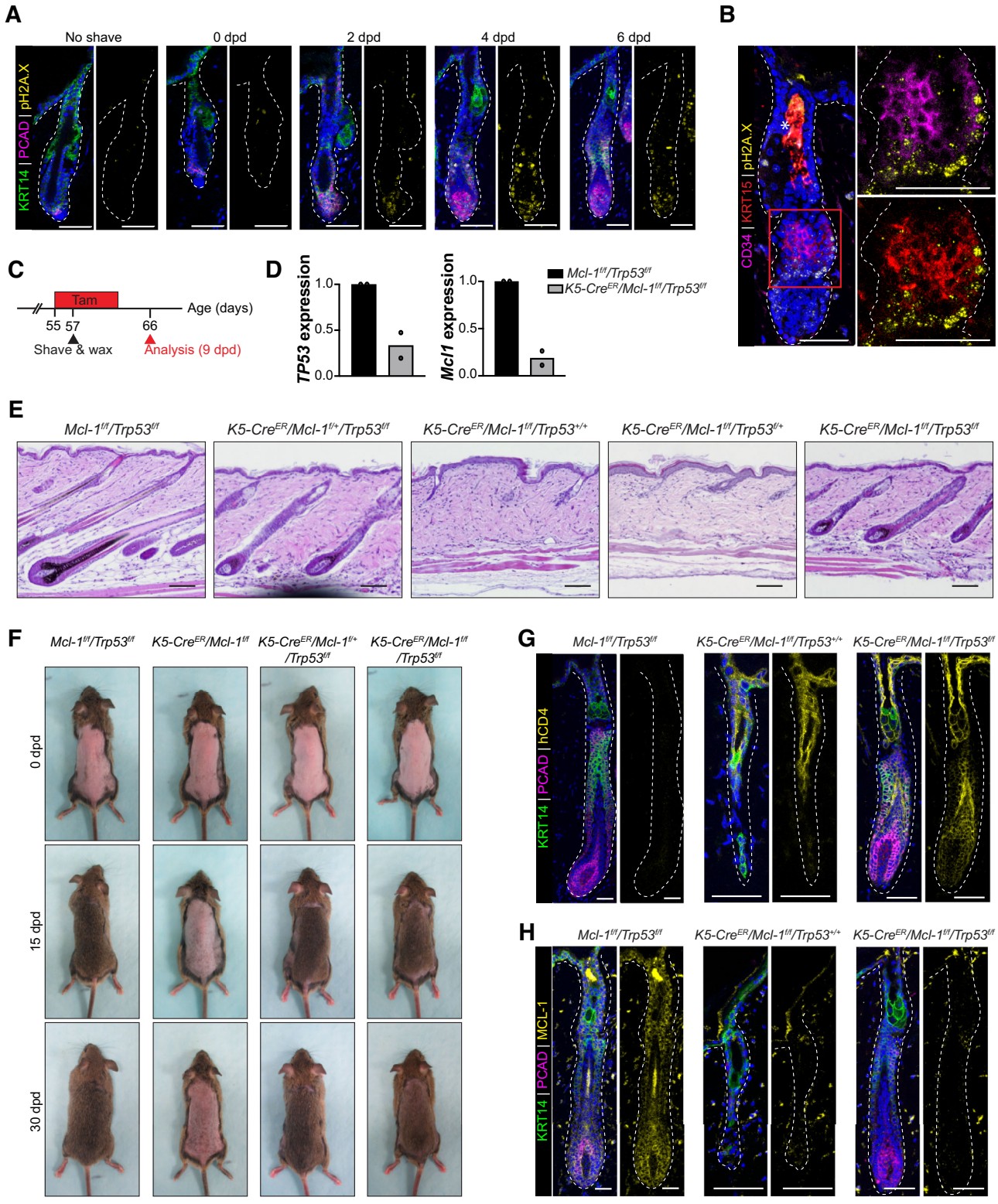

**Fig. 6 | Deletion of *Trp53* rescues hair regeneration defect in *Mcl-1*-deficient mice. A**, **B** Confocal analysis of HFs from FVB mice at P60, sampled at 0, 2, 4, and 6 days post depilation (dpd) to induce hair regeneration. Increased DNA damage (pH2A.X⁺) is evident in HF cells post-depilation (**A**). Basal layer marked by KRT14, hair matrix by PCAD, DAPI as nuclear counterstain. DNA damage observed in a subset of CD34⁺ and KRT15⁺ HFSCs at 4 dpd (**B**). Data represent *n* = 3 mice per time point. Scale bars, 50 µm. **C** Experimental strategy for hair regeneration assay in adult *K5-Cre^ER^/Mcl-1^f/f^/Trp53^fl/f^* mice. *Mcl-1* and *Trp53* deletion was induced via tamoxifen-containing diet between p55-p62. A portion of the dorsal skin was shaved and waxed to induce hair regeneration. Dorsal skin samples were harvested at the indicated timepoints for analysis. **D** qPCR analysis showing

reduced *Mcl-1* and *Trp53* transcript levels in HF cells (CD49f⁺/SCA-1⁻) isolated from *K5-Cre^ER^/Mcl-1^f/f^/Trp53^fl/f^* mice, compared to littermate control. *n* = 2 mice per genotype. **E** Representative H&E-stained histological sections of dorsal skin harvested at 9 dpd. *n* = 6 mice per genotype for each timepoint. Scale bars, 100 µm. **F** Representative images of mice taken at 0, 15, 30 dpd. *n* = 3 mice per genotype for each timepoint. **G**, **H** Confocal microscopy images of HFs from *K5-Cre^ER^/Mcl-1^fl/f^/Trp53^fl/f^* mice and littermate controls at 9 dpd, showing efficient CRE-mediated recombination with upregulation of hCD4 (**G**) and downregulation of MCL-1 (**H**). *n* = 6 mice per genotype for each timepoint. Scale bar, 50 µm. Basal layer is stained by KRT14, hair matrix stained by PCAD and DAPI as a nuclear counterstain.

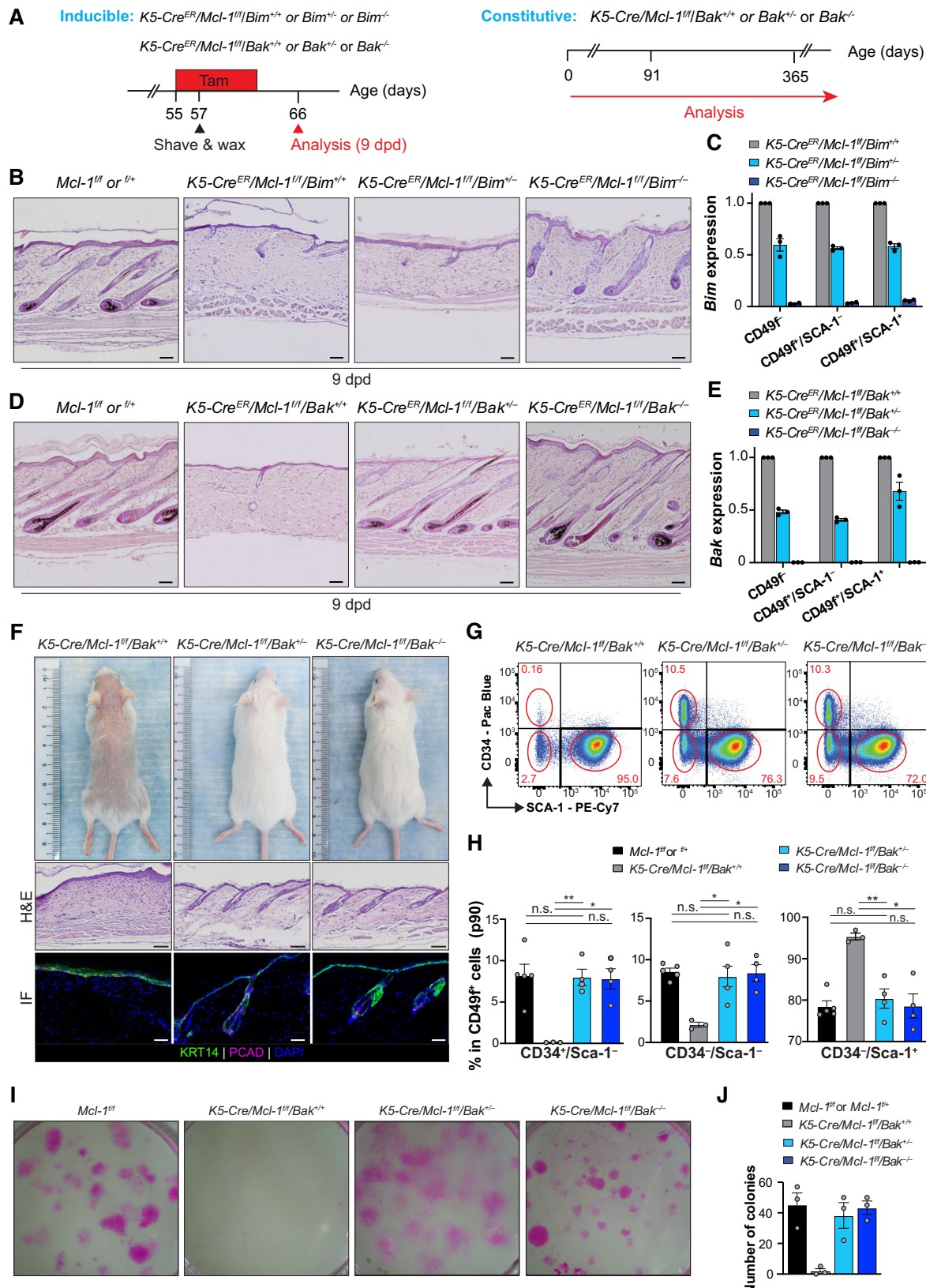

and HFSCs within the skin. Hair loss with the complete destruction of HF structures also occurred when *Mcl-1* gene deletion was acutely induced in adult skin epithelium using a tamoxifen-inducible CRE$^{ERT2}$ system (*K5-Cre$^{ER}$*). Furthermore, our findings from this inducible knockout system highlight the critical role of MCL-1 in the early stages of HF regeneration in adulthood.

Acute deletion of *Mcl-1* following hair depilation inhibited the initiation of HF regeneration, leading to the rapid loss of CD34$^+$ HFSCs and the destruction of HF structures. In contrast, without depilation, quiescent CD34$^+$ HFSCs tolerated *Mcl-1* deletion without adverse effects. Interestingly, while MCL-1 is highly expressed in the IFE, the development and structure of

**Fig. 7 | Rescue of HF defects in *Mcl-1*-deficient mice by concomitant loss of *Bim* or *Bak*. A** Schematics of experimental strategies to assess whether deletion of *Bim* or *Bak* rescues HF defects in the inducible *K5-Cre$^{ER}$/Mcl-1$^{f/f}$* (left panel) and constitutive *K5-Cre/Mcl-1$^{f/f}$* (right panel) models. *K5-Cre$^{ER}$/Mcl-1$^{f/f}$* mice with deletion of single or both alleles of *Bim* or *Bak* at p55 were administered with tamoxifen containing diet to induce *Mcl-1* deletion, followed by depilation. Skin samples were harvested at 9 dpd for analysis. *K5-Cre/Mcl-1$^{f/f}$* mice with single or both *Bak* alleles deleted were generated and analyzed. **B, C** *Bim* deletion partially rescues HF regeneration defects in *K5-Cre$^{ER}$/Mcl-1$^{f/f}$* mice. Representative H&E-stained sections of dorsal skins harvested at 9 dpd. *n* = 6–8 mice per genotype. Scale bars, 100 μm (**B**). qPCR analysis of *Bim* expression in FACS sorted subpopulations. Data are mean ± SEM from 3 mice per genotype (**C**). **D, E** *Bak* deletion completely rescues HF regeneration defects in *K5-Cre$^{ER}$/Mcl-1$^{f/f}$* mice. Representative H&E-stained histological sections of dorsal skins harvested at 9 dpd. *n* = 7–9 mice per genotype. Scale bars, 100 μm (**D**). qPCR analysis of *Bak* expression in FACS sorted subpopulations. Data are mean ± SEM from 3 mice per genotype (**E**). **F** Representative images of *K5-Cre/Mcl-1$^{f/f}$/Bak$^{+/+}$*, *K5-Cre/Mcl-1$^{f/f}$/Bak$^{+/-}$* and *K5-Cre/Mcl-1$^{f/f}$/Bak$^{-/-}$* mice at p90. *n* = 7–8 mice per genotype. Representative H&E-stained sections and confocal images of dorsal skins harvested at p90. Scale bars, 100 μm (H&E), 50 μm (confocal). Basal layer stained by KRT14, hair matrix stained by PCAD and DAPI as nucleus counterstain. **G, H** FACS analysis of CD34 and SCA-1 expression in epithelial cells from mice of indicated genotypes at p90 (**G**). Bar graph shows percentage of each subpopulation in CD49f$^+$ epithelium (**H**). Data are mean ± SEM, *n* = 3–5 per genotype. \**p* < 0.05; \*\**p* < 0.01; n.s., non-significant; Student's *t*-test. **I, J** In vitro clonogenic assay of CD34$^+$ HF stem cells from mice of indicated genotypes at p21, grown on 3T3 feeder cells for 14 days. **J** Colony counts for each genotype, mean ± SEM for *n* = 3 independent experiments. \*\**p* < 0.01; Student's *t*-test.

*Mcl-1*-deficient IFE remained largely intact, with no detectable abnormalities in skin barrier and stratification, and inflammation response. Efficiently and specifically targeting all stem cell subsets within the heterogeneous HFSC compartment remains a significant technical challenge. While *Lgr5* has been reported to serve as a genetic marker for a subset of HFSCs, there is compelling evidence suggesting that *Lgr5$^+$* HFSCs are not positioned at the apex of the HFSC hierarchy[65]. Instead, more quiescent CD34$^+$LGR5$^-$ HFSCs can give rise to LGR5$^+$ cells and compensate their loss during hair regeneration[66]. We found that *Mcl-1* deletion in *Lgr5$^+$* HFSCs did not significantly impair hair regeneration after depilation, which highlights the complexity of the HFSC niche and underscore the need for further investigation in the future to pinpoint the specific HFSC populations reliant on MCL-1 for survival and function. Collectively, these findings underscore the specific role of MCL-1 in activated HFSCs and suggest that epithelial cells in IFE and quiescent stem cells in HFs have distinct pro-survival dependencies. This phenomenon is consistent with studies in other tissues, such as T cell development, where different cell lineages and stages rely on distinct pro-survival BCL-2 family members: MCL-1 is required for DN2 thymocytes, BCL-XL or DP T cells and BCL-2 for mature T cells[32,67].

To delineate the molecular and cellular landscapes during the initial phases of HF regeneration, we conducted scRNA-seq profiling on dorsal skin samples across six time points, spanning early depilation-induced anagen stage from as early as 4 h and up to 6 dpd. Our data unveil the quick response of both HG and OB cells in preparation for HF regeneration, even before cell division occurred. A distinct subset of OB cells with a defined gene signature emerged as early as 4 h post-depilation, preceding the formation of SHH expressing TAC progenitors. These TAC progenitors play a key role in activating quiescent stem cells, prompting them to re-enter the cell cycle and contribute to subsequent HF growth. The most significant changes in gene expression within the OB cells occurred from 2 dpd, coinciding with the activation of quiescent cells, the initiation of proliferation, and generation of progenitor or more committed cell subsets. Genes associated with cell cycle and DNA replication were highly upregulated at this stage. Simultaneously, the DNA repair and P53 pathways were activated, likely serving as a critical mechanism coordinating cell proliferation and apoptosis.

The activation of the WNT/β catenin pathway was shown to induce the accumulation of acetylated P53 and activation of the P53 pathway in both HG and activated CD34$^+$ HFSCs during early anagen stages[68]. Interestingly, while BCL-2 deficient mice did not exhibit obvious hair growth defects under normal physiological conditions, a delay in depilation-induced hair regeneration was observed in BCL-2 null mice[23], and this phenotype was rescued by the concomitant absence of *Trp53*[69]. Remarkably, *Trp53* deletion rescued the hair regeneration defects caused by *Mcl-1* deficiency upon depilation, highlighting the critical interaction between these proteins in balancing proliferation and cell death. Proliferation stress, experienced during rapid cell division, places significant demands on DNA replication, repair, and genomic stability, increasing the risk of DNA damage[70]. In response, the P53 pathway becomes activated to either promote DNA repair or trigger apoptosis if damage is irreparable[71]. Our findings show that MCL-1 protects HFSCs from P53-mediated apoptosis during proliferation stress, ensuring their survival and enabling successful hair regeneration. The interplay between P53 and MCL-1 provides a framework for understanding how proliferation stress and apoptosis are balanced during tissue regeneration, with potential relevance to stem cell regulation across different tissues. It has been previously shown that P53-mediated DNA damage response is triggered when quiescent lung stem cells are activated and undergo a transient state for regeneration while it remains to be determined whether MCL-1 also plays an important role in the process[72].

The interplay between different pro-survival and pro-apoptotic members of the BCL-2 protein family controls cell fate. Pro-survival BCL-2 family members regulate cell survival by sequestering BH3-only proteins and multi-BH domain effectors, BAX, BAK, and BOK[18]. Unlike other pro-survival BCL-2 members, MCL-1 exhibits both anti-apoptotic and apoptosis unrelated functions. To address whether the HF phenotypes associated with *Mcl-1* deficiency arise from its anti-apoptotic function, we deleted pro-apoptotic BCL-2 family members that function upstream and downstream of MCL-1. Remarkably, deletion of even a single *Bak* allele completely prevented the hair regeneration defects and the loss of HFSCs caused by *Mcl-1* deficiency, indicating that BAK plays a crucial role in promoting apoptosis in HFs, with MCL-1 primarily functioning to restrain BAK. However, whether MCL-1 predominantly inhibits one of BAK, BAX or BOK, or a combination of these proteins remains an open question. Interestingly, a recent study in the brain showed that conditional knockout of *Mcl-1* led to progressive white matter degeneration, myelination defects, and the loss of mature oligodendrocyte populations, and deletion of a single *Bak* allele prevented these defects[73]. The BH3-only proteins PUMA, NOXA and BIM have been shown to physically interact with MCL-1 thereby unleashing BAK and BAX from restraint to promote apoptosis[74–76]. Our data indicate that *Bim* deletion only partially rescues hair regeneration defects caused by *Mcl-1* deficiency, suggesting that other BH3-only protein(s) alongside BIM act upstream of MCL-1 to initiate the apoptotic cascade in HFs. Collectively, our data reveal that BIM, alongside other pro-apoptotic BH3-only proteins, interacts with MCL-1 in HFs, to unleash the pro-apoptotic effector BAK to drive apoptosis in HFs.

The expression of MCL-1 can be regulated through multiple mechanisms, including transcriptional, translational, and post-translational controls. Our findings suggest that, in both cultured

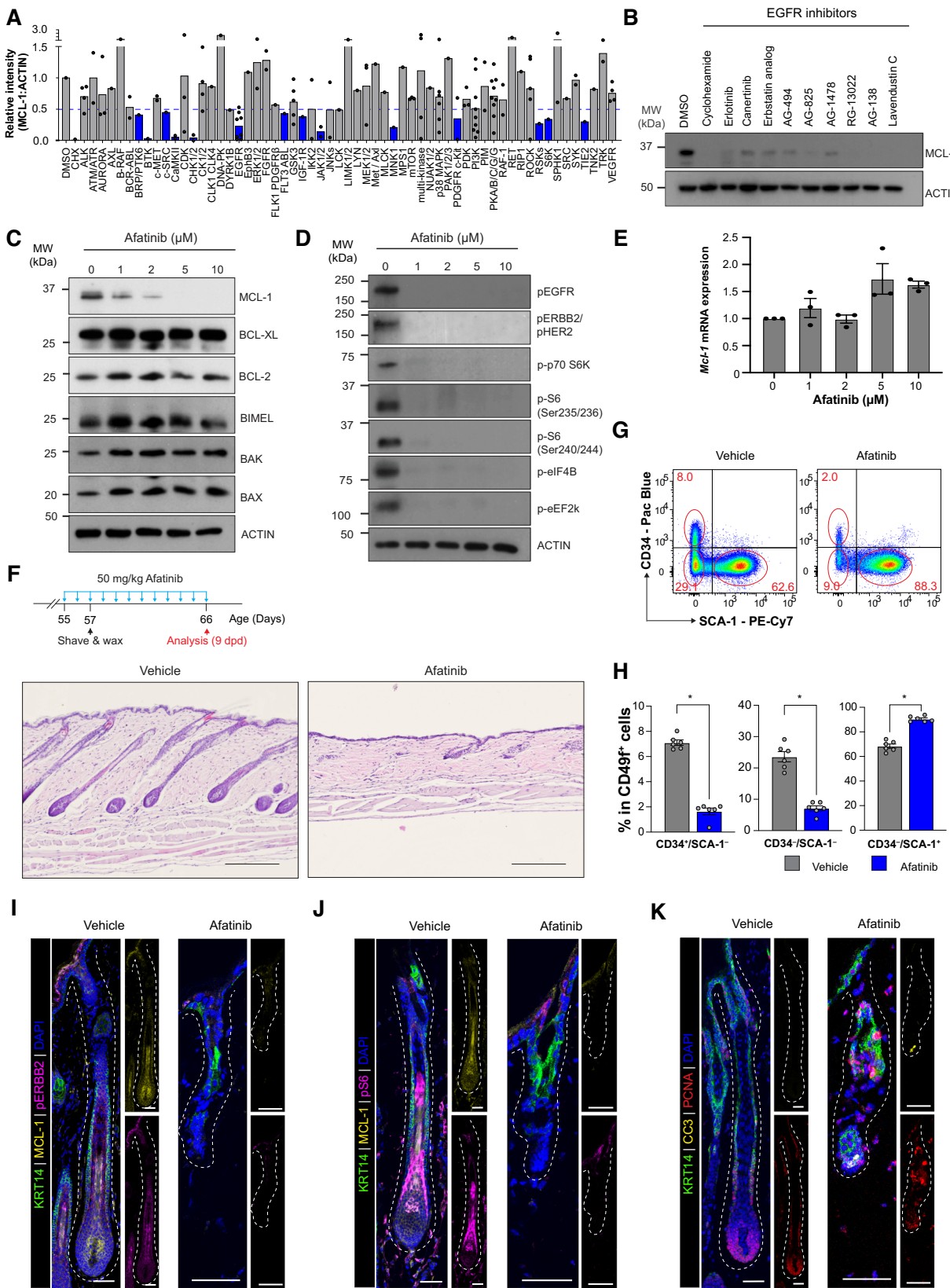

primary keratinocytes and HFs, MCL-1 expression is subjected to post-transcriptional regulation by the ERBB signaling pathway. The cyclical nature of hair growth is orchestrated by a complex regulatory network involving various growth factors, including EGF[77,78]. While EGFR-deficient mice do not survive embryonic development, mutant strains with EGFR signaling defects survive for up to 3 weeks

post-birth, displaying impaired skin and hair growth[79,80]. Moreover, the overexpression of a dominant negative mutant of EGFR in K5-expressing cells, including the skin epidermis and ORS, did not perturb HF morphogenesis but caused progressive hair loss[81]. This aligns with observations from our constitutive *Mcl-1* knockout mouse model, although the impact on HFSCs was not addressed in the

**Fig. 8 | Inhibition of ERBB signaling blocks MCL-1 protein expression and hair regeneration. A** Kinase inhibitor screen in primary keratinocytes identifies pathways that modulate MCL-1 protein levels. Cells from newborn FVB/N pups were treated with 2 μM of each inhibitor from a panel of 154 kinase inhibitors for 24 h, followed by Western blotting for MCL-1 expression. The intensities of MCL-1 and ACTIN were analysed using Image J. The ratio of MCL-1 vs Actin in inhibitor-treated cells with normalization to DMSO treated control cells were plotted as a bar graph. Inhibitors were grouped based on the kinase targeted. Groups highlighted in blue show more than a 50% reduction of MCL-1 levels on average (MCL-1 *vs* Actin ≤ 0.5). **B** Validation of ERBB inhibitors from (**A**) shows significant reduction in MCL-1 protein levels after 24 h of treatment ($n = 2$ independent experiments). **C** Western blot of keratinocytes treated with various afatinib concentrations for 24 h demonstrates MCL-1 reduction without affecting other BCL-2 family proteins ($n = 3$ independent experiments). **D** Afatinib inhibits phosphorylation of protein translation regulators in primary keratinocytes. Western blot of p-EGFR, p-ERBB2, p-P70 S6K, p-S6, p-eIF4B, and p-eEF2K ($n = 2$ independent experiments). **E** qPCR analysis of afatinib-treated keratinocytes reveals *Mcl-1* transcript levels remain stable. Data are mean ± SEM ($n = 3$ independent experiments). **F** H&E-stained sections of dorsal skin from wildtype FVB/N mice treated with 50 mg/kg afatinib or vehicle followed by depilation show impaired hair regeneration at 9 dpd ($n = 9$ mice per treatment group). Scale bars, 50 μm. **G, H** FACS plots and bar graphs indicate changes in CD34 and SCA-1 populations in CD49f⁺ epithelial cells from afatinib- or vehicle-treated mice at 9 dpd. Data are mean ± SEM. ****$p < 0.0001$; **$p < 0.01$; *$p < 0.05$; Student's *t*-test. **I–K** Confocal images of HFs from afatinib- or vehicle-treated mice at 9 dpd. **I** MCL-1 and p-ERBB2 expression; **J** MCL-1 and p-S6 expression; **K** Proliferating (PCNA⁺) and apoptotic (CC3⁺) cells. Scale bars, 50 μm. Basal layer stained by KRT14 and DAPI as nuclear counterstain ($n = 6$ mice per treatment arm).

previous study. Additionally, transgenic mice overexpressing EGF demonstrated prolonged anagen phase and a failure to undergo catagen due to a reduced apoptosis[78]. Consistent with this, we noted a downregulation of ERBB signaling in the apoptotic strand during catagen, which was associated with decreased levels of MCL-1 and p-S6 proteins. These findings suggest that ERBB signaling modulates MCL-1 expression through translational control mechanisms, particularly during periods of heightened apoptosis and regression in the hair cycle, while it remains to be determined which ligands and receptors may play a dominant role in regulating MCL-1 protein levels in different subsets of HF cells. Moreover, afatinib and erlotinib treatment in mice reduced MCL-1 protein levels and prevented the initiation of HF regeneration following depilation. Together, these results underscore the pivotal role of ERBB signaling in hair regeneration and HFSC maintenance by regulating the protein translation of the pro-survival *Mcl-1* gene. Importantly, our scRNA-seq profiling data suggest that multiple ligands and ERBB receptors are expressed in skin cells, posing a challenge in identifying which ligands and receptors play key roles in different skin cell lineages.

In conclusion, this study sheds light on the crucial role of the pro-survival BCL-2 family member, MCL-1, in regulating HF regeneration during adulthood. Our findings demonstrate that MCL-1 is essential for the survival of activated HFSCs during hair regeneration but dispensable for quiescent HFSCs. We reveal a crucial interplay between pro-survival and pro-apoptotic BCL-2 family members, notably BAK, in controlling cell fate during HF regeneration. Importantly, our data highlight the interplay between P53 and MCL-1, offering new insights into balancing proliferation stress and apoptosis during tissue regeneration, with *Trp53* deletion rescuing the hair regeneration defects caused by *Mcl-1* loss. Furthermore, we mapped the molecular and cellular dynamics during the early stages of HF regeneration, providing a comprehensive skin cell atlas and identifying key gene expression programs that govern this process. The ERBB signaling pathway also emerged as a regulator of HF regeneration through its promotion of MCL-1 expression. Together, this study advances our understanding of the molecular mechanisms underlying HF regeneration and offers new insights into how stem cell survival and tissue regeneration are orchestrated. These findings may have broader implications for controlling the survival of stem and progenitor cells in tissue regeneration and cancer expansion.

## Methods
### Mice
*Mcl-1^{f/f}* (FVB/C57BL/6 mixed background)[82], *Bim^{-/-}* (C57BL/6 background)[55], *Bak^{-/-}* (C57BL/6 background)[58], *TrpS3^{f/f}* (mixed background)[83], and *Rosa26^{mTmG}*[40] (C57BL/6 background) have been described previously. *Keratin 5 (K5)-Cre*[84], (C57BL/6 background), *K5-Cre^{ER}* (C57BL/6 background) and *Lgr5-T2A-Cre^{ER}*[47] were gifts from Dr J. Takeda (Osaka University, Japan), Dr B. Hogan (Duke University Medical Center) and Dr Nick Barker (A*STAR, Singapore), respectively. The *K5-Cre/Mcl-1, K5-Cre/Mcl-1/Bak* and *K5-Cre^{ER}/Mcl-1* lines were backcrossed with FVB wild types for more than 10 generations. A mix of both male and female mice were used for all experiments and no gender-based differences were observed. All animals used in this study were bred and housed in individually ventilated cages in specific pathogen-free holdings with temperature and light control (12 h light and dark cycle).

*K5-Cre^{ER}* mediated deletion of floxed *Mcl-1* or *Trp53* alleles was carried out by placing mice on an irradiated tamoxifen diet (Envigo). To inhibit ERBB signaling, afatinib (50 mg/kg body weight/day) and erlotinib (25 mg/kg body weight/day) were administered by oral gavage at p55 until the end of the experiment as indicated. To induce hair regeneration, the dorsal skin was shaved, and a wax strip was used to completely remove the hair from the dorsal skin. All animal experiments were carried out under the approval of and guidelines of the SingHealth Institute Animal Care and Use Committee, and WEHI institutional guidelines in approval by the WEHI Animal Ethics Committee.

### Histology and immunofluorescent staining
Skin was fixed in 4% paraformaldehyde for 24 h at room temperature, embedded in paraffin, sectioned at 8 μm and stained with hematoxylin and eosin (H&E). For immunofluorescent staining, slides were re-hydrated through a descending series of ethanol to water. Antigen retrieval was performed using a pressure cooker for 10 min at the highest pressure in sodium citrate solution (pH 6). Slides were cooled down on ice and washed in tap water before blocking with PBS containing 0.1% Triton X-100 in PBS, 10% horse serum, and 2% mouse serum on mouse blocking reagent for 30 min. Slides were washed in PBS containing 0.01% Triton X-100 and blocked with antibodies in PBS containing 0.01% Triton X-100 and 1% Horse serum. For immuno-fluorescent staining, the following antibodies were used: anti-MCL-1 (rabbit, clone D2W9E, 1:300), anti-BCL-XL (rabbit, clone 54H6, 1:200), anti-BIM (rabbit, clone C34C5, 1:100), anti-BAK (rabbit, clone D4E4, 1:100), anti-activated (i.e. cleaved) CASPASE-3 (rabbit, clone 5A1E, 1:100), anti-pHER2/pErbB2 (Y1221/1222) (rabbit, clone 6B12, 1:100), anti-phospho-histone H2A.X (Ser139) (rabbit, clone 20E3, 1:200), anti-P-S6 ribosomal protein (S240/244) (rabbit, clone D68F8, 1:100), anti-LEF1 (rabbit, clone C12A5, 1:100), anti-PCNA (mouse, clone PC10, 1:100), and anti-E-Cadherin (Rabbit, clone 24E10, 1:200) antibodies from Cell Signaling; anti-mouse P-CADHERIN (goat, clone AF761, 1:400) antibody from R&D systems; anti-mouse CD34 (rat, clone RAM34, 1:50) antibody from eBiosciences; anti-pan-cytokeratin (AE-13) (mouse, clone sc-57012, 1:300), anti-CLAUDIN-1 (mouse, clone A-9, 1:100), as well as anti-KRT15 (mouse, clone LHK15, 1:200) antibodies from Santa Cruz; anti-GATA3 (mouse, clone L50-823, 1:50) antibody from BD Pharmingen; anti-FILAGGRIN antibody (rabbit, 1:100) from BioLegend; anti-DESMOPLAKIN (rabbit, 1:100) and anti-PLECTIN (rat, 1:100) antibodies from BiCell; anti-GFP (chicken, 1:300), anti-LORICRIN (rabbit, 1:100), anti-CD4 (Rabbit, EPR19514, 1:200), and anti-CD8 (rabbit, EPR21769, 1:200) antibodies from Abcam; anti-RFP antibody

(rabbit, 1:200) from Rockland; anti-KRT14 (rat, 1:1000), anti-KRT-10 (rabbit, 1:1000) and anti-KRT6 (rat, 1:1000) antibodies from Chen Ting Lab. Images were acquired with a Leica TCS SP8 confocal microscope. Microscopy data were analyzed using ImageJ. RGB images were assembled and labeled with Adobe Illustrator.

### Skin and mammary cell preparation from adult mammary tissue for fluorescence-activated cell sorting (FACS) analysis

To sort for skin cells in telogen stage, subcutaneous fat was removed with a scalpel. Skin was cut into small pieces and floated, dermis side down on 0.25% trypsin at 37 °C for 30 min. Skin epithelial cell suspensions were obtained by gentle scrapping with a scalpel. 5% FBS was added to inactivate trypsin, and single cell suspensions were obtained by passing through a 40 μm strainer. For isolating skin cells in anagen stage, skin was cut into small pieces and digested in 300 U/mL collagenase (Sigma) and 100 U/mL hyaluronidase (Sigma) in HBSS (GIBCO) at 37 °C for 1–2 h. The digested tissues were centrifuged and resuspended in 0.25% Trypsin (GIBCO) and incubated at 37 °C for 20 min. 5% FBS was added to inactivate trypsin, and single cell suspensions were obtained by passing through a 40 μm strainer. Mammary glands from adult virgin female mice were collected. Single-cell suspensions were prepared essentially as previously described[85].

### Fluorescence-activated cell sorting (FACS) analysis

Non-specific staining of cells was blocked by incubation with rat γ-globulin (Jackson Laboratories) and anti-CD16/CD32 Fcγ III/II receptor antibody (BD Pharmingen) for 10 min. Antibody incubations were then performed on ice for 30 min. The following antibodies were used: APC/cy7 anti-mouse/rat CD29 (rat, clone HM1-1, Cat. no. 102226, 1:200), Pacific Blue anti-mouse CD24 (rat, clone M1/69, Cat. no. 101820, 1:200), eFluor 450 anti-mouse CD34 (rat, clone RAM34) antibody from eBioscience; PE-Cy7 anti-mouse SCA-1 (Ly-6 A/E) (rat, clone D7), BUV395 anti-human CD49f (integrin α6 chain) (rat, clone GoH3), BV711 anti-human CD4 (mouse, L120 (RUO)), APC anti-mouse CD31 (rat, clone MEC13.3), APC anti-mouse CD45 (rat, clone 30-F11), APC anti-mouse TER119 (rat, clone TER119) antibodies from BD Pharmingen, PE anti-mouse Tspan8 (rat IgG2b clone no. 657909, Cat. no. FAB6524P, 1:75) and APC anti-mouse Tspan8 (Rat IgG2b clone no. 657909, Cat. no. FAB6524A, 1:75) from R&D Systems. To exclude dead cells, cells were resuspended in 0.2 μg/mL 7-AAD (CAYMAN) before analysis. The Lin⁻ cell population was defined by TER119⁻CD31⁻CD45⁻. FACS analysis was performed on a LSRFortessa (Becton Dickinson), while FACS sorting was performed on an AriaII (Becton Dickinson). FACS data were analyzed using FlowJo software (TreeStar).

### scRNA-seq data analysis

The 10x Genomics Chromium sequencing data were processed using Cell Ranger 6.1.2. The filtered count matrix from Cell Ranger was used for analysis. An average of ~22 million reads and ~10,000 cells were obtained for each sample. Quality control was first performed before the downstream analysis. Cells with detected genes fewer than 500 and with mitochondria read percentage greater than 10% were removed for each sample, as these cells are likely to be damaged cells and of low quality. Cells with detected genes greater than 6000 or 7000 were removed for different samples, as these cells are likely to be doublets or outliers.

After quality control, Seurat (v4.1.1)[17] standard pipeline was performed on each sample. The scRNA-seq data was first normalized by using the NormalizeData function. Then, top 2000 highly variable genes (HVGs) were selected by FindVariableFeatures function. Normalized expression data of the 2000 HVGs were scaled to have a mean of 0 and a variance of 1 by ScaleData function. For dimension reduction, the principal component analysis (PCA) was performed on HVGs using RunPCA function, and Uniform manifold approximation and projection (UMAP) was performed on the first 30 PCs. The doublet

prediction step was added to the standard pipeline to predict the potential doublets using scDblFinder (v1.10.0)[86] on the PCA results. Clusters were identified by FindNeighbors function and FindClusters function with an appropriate resolution value. Cell type annotation was done by using known marker genes from previous studies[48,49]. The annotated HF cells were then kept without doublets, and the same Seurat pipeline was performed on HF cells of each sample.

Integration analysis of singlets of all the samples (0 h, 4 h, 16 h, 2 d, 4 d and 6 d) was performed using the Seurat anchor-based integration pipeline. Normalization and HVG selection were first performed on each sample. Anchor genes among samples were identified by using the SelectIntegrationFeatures function. Anchors were identified by FindIntegrationAnchors function using the anchor genes. The IntegrateData function was then used to integrate all samples together taking the anchors as input. The integrated assay was then scaled. PCA and UMAP were performed to obtain the reduced dimensions. Clusters were identified with resolution of 0.2. Cell types were annotated using known marker genes. HF cells from integration analysis were extracted, and the Seurat integration pipeline was performed again on HF cells only. Positive marker genes of each cell type were identified by FindAllMarkers function with log2 fold change threshold of 0.25 on the RNA assay. RNA velocity analysis was performed after integration analysis of HF cells. Velocyto[87] was used to obtain the spliced and unspliced count for each sample. scVelo[88] was used to estimate RNA velocity.

Pseudobulk samples were constructed by aggregating counts of each cell type of each sample for pseudotime course analysis. Pseudotime course analysis were performed for each cell type following the edgeR time course analysis workflow[89]. The quasi-likelihood method in edgeR[90] was used to identify the DE genes along pseudotime course, and KEGG analysis was performed using edgeR's kegga function in edgeR on the up and down-regulated DE genes from the linear-trend coefficient.

Hair germ (HG) cells were identified from an integrated Seurat analysis of HF cells from the 0 h, 4 h and 16 h samples. Read counts were extracted for the HG cells from these three samples and a differential expression analysis taking each HG cell as a sample was undertaken using the new quasi-likelihood pipeline in edgeR v4[91]. DE genes were obtained for 4 h vs 0 h, for 16 h vs 0 h and for the average of 4 h and 16 h vs 0 h. KEGG analysis of the gene lists was performed using edgeR's kegga function.

For DE analysis of hair germ cells, the hair germ cells of 0 h, 4 h and 16 h samples were obtained from Seurat integration analysis of HF cells of the three samples. The 4 h and 16 h samples were merged and assigned to the same group 4h_16h. The single cell level quasi-likelihood method in edgeR was used to identify the DE genes between 4h_16h and 0 h taking each cell as a sample. KEGG analysis was performed by using kegga function.

### In vitro colony formation assays

FACS sorted Lin⁻/CD49f⁺/CD34⁺/SCA-1⁻ cells from *K5-Cre/Mcl-1^(f/f)*, *K5-Cre/Mcl-1^(f/f)/Bak^(-/-)*, *K5-Cre/Mcl-1^(f/f)/Bak^(+/-)* and *Mcl-1^(f/f)* or *Mcl-1^(f/+)* mice were obtained and cultured on a feeder layer of mitomycin C-treated NIH3T3 cells at 37 °C, 5% CO₂. Keratinocytes were cultured in keratinocyte serum free medium (GIBCO) for 14days, then fixed and stained with 1% (wt/vol) Rhodamine B (Sigma). Numbers of colonies from scanned images of plates were counted using ImageJ.

### Western blotting

FACS sorted skin cell sub-populations and cultured keratinocytes were directly lysed on ice with lysis buffer (50 mM Tris-HCl (pH 7.6), 150 mM NaCl, 1% Triton-X100) containing 1x complete mini protease inhibitor cocktail (Roche). The following primary antibodies were used for Western blotting: anti-MCL-1 (rabbit, clone D2W9E, 1:1000), anti-BCL-XL (rabbit, clone 54H6, 1:1000), anti-BCL-W (rabbit, clone 31H4, 1:500), anti-BIM (rabbit, clone C34C5, 1:1000), anti-BAK (rabbit, clone D4E4, 1:1000), anti-p-EGFR (Y1068) (rabbit, clone D7A5, 1:500),

anti-pHER2/pErbB2 (Y1221/1222) (rabbit, clone6B12, 1:100), anti-P-S6 ribosomal protein (S235/236) (rabbit, clone D57.2.2E), anti-P-S6 ribosomal protein (S240/244) (rabbit, clone D68F8, 1:100), anti-p-eIF4B (S422) (rabbit, 1:1000), anti-p-eEF2K (S366) (rabbit, 1:1000), anti-p-P70 S6 kinase (Thr389) (rabbit, clone 108D2, 1:1000), anti-P70 S6 kinase (rabbit, clone 49D7, 1:1000) antibodies from Cell Signaling; anti-BCL-2 (mouse, clone C-2, 1:1000) antibody from SantaCruz; anti-BAX (rat, gift from David Huang (WEHI, 1:1000). The following secondary antibodies were used: anti-mouse IgG (H + L)-HRP (goat, 1:10000), anti-rabbit IgG (H + L)-HRP (goat, 1:10000) antibodies from Bio-Rad; anti-rat IgG (H + L)-HRP (goat, 1:30000) antibody from Bethyl Laboratories. The chemiluminescence Immobilon® Forte Western HRP substrate (Merk-Millipore) was used to detect proteins by Western blotting.

## Cell culture experiments

To establish cell lines from the skin of FVB/N mice, dorsal skin was removed from newborn mice and placed in 0.4 mg/mL dispase in PBS for 1 h at 37 °C. The epidermis was separated from the dermis and digested in 0.25% trypsin (GIBCO) at 37°C for 10 min. 5% chelated FBS was added to inactivate trypsin and skin epithelial cells were collected by centrifugation and cultured in keratinocyte serum-free medium (GIBCO). All cell lines were cultured at 37 °C with 5% $CO_2$. For experiments using the kinase screening library, keratinocytes were cultured with 2 μM of each compound (CAYMAN; 96-well; item 10505; batch: 0577971) according to the ascending list starting at well A1 to H12 of plate 1 followed by plate 2, irrespective of the drug target. As controls, keratinocytes were also cultured with DMSO and cycloheximide (CHX) (100 μg/mL), respectively. As MCL-1 protein has a short half-life, keratinocytes treated with CHX served as a positive control. Gel images were scanned, and the intensities of signals were calculated using ImageJ. The MCL-1 versus Actin intensity ratio for each treatment was normalized to DMSO treated cells within each gel. To verify the kinase screening library, keratinocytes were cultured with 2 μM of compounds from the library that target the EGFR/ERBB2 signaling pathway. Keratinocytes also were cultured for 24 h with an independent dual inhibitor targeting EGFR/ERBB2, afatinib, at the indicated concentrations at 37 °C.

## RT-qPCR

FACS sorted skin cell subpopulations were obtained as described above and total RNA was prepared by using the RNeasy Micro kit (Qiagen). Equal amounts of RNA were added to the reverse-transcriptase reaction mix. Real-time PCR was conducted using a CFX96TM Real-Time system (Bio-Rad) with PowerUP SYBR Green master mix (AppliedBiosystems). Relative expression of *Mcl-1, Bim, Bak, Beta-actin* mRNA was calculated using the delta-delta Ct ($2^{-\Delta\Delta Ct}$) method. Sequences of primers used for the real-time PCR can be found in our supplementary methods.

## Statistical analysis

Data are presented as the mean value ± standard error of mean (S.E.M.). All graphs were prepared in GraphPad Prism. Statistical analyses were performed using the unpaired two-tailed Student's *t*-test in GraphPad Prism, with $p < 0.05$ considered significant.

## Reporting summary

Further information on research design is available in the Nature Portfolio Reporting Summary linked to this article.

## Data availability

All the data generated and analyzed during this study are included in this published article and its supplementary information files. The scRNA-seq data generated in this study have been deposited in the Gene Expression Omnibus (GEO) depository under the accession code: GSE256251. The source data are provided with this article. Source data are provided with this paper.

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

## Acknowledgements

We are grateful to Z.M. Lim and N.H. Hamadee for assistance in FACS analysis/sorting and histology. We thank Dr. M.M.D. Varela and H.L. Chung at SingHealth Animal Vivarium. This work was supported by MOE Tier 2 (MOE2017-T2-2-047 and MOE2019-T2-2-006). F.N.Y. and H.S.C. were supported by Victorian Cancer Agency Mid-Career Research Fellowship (MCRF22013) and NMRC OF-YIRG (MOH-000546), respectively; P.S. was supported by the Biomedical Research Council (BMRC) and Central Research Fund (CRF); N.B. was supported by the Agency for Science, Technology and Research (A*STAR); G.J.L. was supported by the Breast Cancer Research Foundation and a NHMRC Fellowship (#1078730, 1175960, 2026004); A.S. was supported by NHMRC Investigator grant (#2007887); G.K.S. was supported by NHMRC Fellowship (#1058892) and Investigator Grant (#2025645); J.E.V. were supported by NHMRC Fellowships (#1037230, 1102742); Y.C. was supported by MRFF Investigator Grant (#1176199).

## Author contributions

N.Y.F. and H.S.C. designed the study. H.S.C., S.H.H., M.Z., G.G.L., S.N., F.G., K.M., F.C.J., J.Y. and J.S.L.T. performed experiments. J.C., Y.C. and G.K.S. performed bioinformatic analysis of scRNA-seq data. N.Y.F., H.S.C., G.J.L, A.S., J.E.V. P.S., T.C. and N.B. carried out interpretation of data. N.Y.F. and H.S.C. wrote the manuscript.

## Competing interests

The authors declare no competing interests.

## Additional information

[1]Cancer and Stem Cell Biology Program, Duke-NUS Medical School, Singapore, Singapore. [2]Bioinformatics Division, The Walter and Eliza Hall Institute of Medical Research, Parkville, VIC, Australia. [3]Cancer Biology and Stem Cells Division, The Walter and Eliza Hall Institute of Medical Research, Parkville, VIC, Australia. [4]Department of Medical Biology, The University of Melbourne, Parkville, VIC, Australia. [5]A*STAR Skin Research Labs (A*SRL), Agency for Science Technology and Research (A*STAR), Singapore, Singapore. [6]Genome Institute of Singapore, Agency for Science Technology and Research (A*STAR), Singapore, Singapore. [7]Institute of Molecular and Cell Biology, Agency for Science, Technology and Research (A*STAR), Singapore, Singapore. [8]Department of Physiology, Yong Loo Lin School of Medicine, National University of Singapore, Singapore, Singapore. [9]School of Mathematics and Statistics, The University of Melbourne, Parkville, VIC, Australia. [10]The Royal Melbourne Hospital, Parkville, VIC, Australia. [11]Peter MacCallum Cancer Centre, Melbourne, VIC, Australia. [12]Blood Cells and Blood Cancer Division, The Walter and Eliza Hall Institute of Medical Research, Parkville, VIC, Australia. [13]National Institute of Biological Sciences, Beijing, China. ✉e-mail: huisan.chin@duke-nus.edu.sg; fu@wehi.edu.au

