## [Transparent Peer Review file · Nature Communications]

MCL-1 safeguards activated hair follicle stem cells to enable adult hair regeneration

Corresponding Author: Professor Nai Yang Fu

Version 0:

Reviewer comments:

Reviewer #1

(Remarks to the Author)

In this manuscript, Chin et al. demonstrate that the anti-apoptotic protein MCL1 is required for hair follicle stem cell (HFSC) survival. Conditional loss of Mcl1 from skin epithelium impaired acute hair regeneration and led to progressive hair loss, depleting HFSCs once activated in both contexts. A kinase screen identified ERBB signaling as an upstream regulator of MCL1 in keratinocytes with the potential to affect hair regeneration. Downstream, parallel deletion of BCL2 family members Bim and Bak partially and fully rescued HFSC number and hair regeneration defects, respectively.

In general, the manuscript is well executed, with reasonably comprehensive phenotyping and pharmacological and genetic phenocopying and rescue experiments lending credence to the ERBB - MCL 1-Bim-Bak pathway mediating survival decisions of HFSCs. However, some major concerns should be addressed to bolster the manuscript (below).

Major concerns:

- 1) Cre-mediated conditional deletion was achieved using a promoter expressed in all skin epithelium, leaving open the possibility that any observed hair defects are secondary. In the absence of a HFSC-specific driver, the authors should be sure that Mcl1 loss does not result in barrier breach, immune invasion, or any other abnormality that can be assigned to the interfollicular epidermis. Is any effect (e.g. proliferation, stratification, etc.) observed in epidermal stem cells (EpdSCs)?
- 2) It is unclear why activated HFSCs would require MCL 1 mediated protection. The authors reference the "stress" associated with stem cell activation, but little is done empirically to show how might interface with these pathways. Do proliferating HFSCs experience DNA damage, or any of the stress pathways identified in the scRNAseq? Any work to this end could offer real insight into the contextual regulators of apoptotic processes, significantly improving this manuscript.
- 3) Analyses of scRNA-sequencing from all cell types derived from depilation-induced skin do not fit well with the rest of the manuscript. The comparisons made between cell types and across time points do not yield information relevant to the author's major conclusions or state somewhat obvious points, such as the upregulation of cell cycle-related genes following stem cell activation. It may be better utilized to identify potential HFSCs stressors which would necessitate Mcl1 anti-apoptotic activity during proliferation. Finally, the data set's potential utility in identifying local sources of ERBB ligands or to dissect the dynamics of key pathway components during stem cell activation are underdeveloped and should be improved.
- 4) The use of afatinib, a tyrosine kinase inhibitor with relative HER2/4 specificity, is relied on too heavily to substantiate claims that ERBB regulates MCL1 expression and blocks hair regeneration. The finding that afatinib broadly impacts the activation of protein translational machinery, in particular, creates some doubt that these effects are solely the result of ERBB inhibition. These data should be supplemented with a more thorough, unbiased examination of its potential off-target effects and more specific (e.g. genetic) evidence, at least in cultured keratinocytes, that ERBB signaling is responsible for regulating MCL1 expression. Rescue experiments are needed to demonstrate that any effects on keratinocyte apoptosis are specific to MCL1.

Minor points:

A) In figure 1 the authors refer to "early" anagen which is conventionally when HFSCs proliferate (anagen II-III); however, the representative images shown are mid-late anagen (at least anagen IV). Data from earlier anagen should be included for

Mcl1, Bclxl, Bak, Bim. For accurate staging please see: Müller-Röver, S. et al. A Comprehensive Guide for the Accurate Classification of Murine Hair Follicles in Distinct Hair Cycle Stages. *Journal of Investigative Dermatology* 117, 3-15 (2001).

B) Do the authors have any insight into why Mcl1 loss has no effect during development? Or why only the HFSC population is affected and not EpdSCs (assuming this is the casual phenotype)? Elaboration in the discussion would be useful.

C) Quantification of phenotypes graphically or definitive statement of percentage of phenotypic animals (eg with greater than 50% hair loss) under each condition should be added in the text.

D) Figure 3A,B is confusing to interpret. Why not present percentage of each cell type (HFSC, IFE, other) of the total, instead of % Cd49f against the other 2 markers?

E) Typo in line 107 ("depletio")

F) Data regarding loss of Cd34+ HFSCs, especially with respect to the depilation experiment, needs some clarification. Is it truly loss of the HFSCs in the Mcl1 conditional knockout within 3 days post depilation, or loss of the CD34 marker? Additional markers of HFSC identity should be investigated (K24, Nfatc1, Sox9, etc.)

G) Figure 6D and 6E references are switched in text (page 15-16).

Reviewer #2

(Remarks to the Author)

The current article by the group of Dr. Naiyang Fu and collaborators uncovers an essential role of the pro-survival protein MCL-1 in regulating HF development and adult hair regeneration. PI and collaborators utilize mouse genetics approach to demonstrate that MCL-1 is critical for the survival and function of activated HF stem cells (HFSCs) during hair regeneration. They demonstrate that constitutive deletion of Mcl-1 resulted in gradual hair loss and the eventual elimination of HFSCs.

Results suggest that MCL-1 is dispensable for quiescent HFSCs, but its acute deletion during the activation phase results in rapid HFSC loss and a complete block in hair regrowth in adult mice. Using single-cell RNA-sequencing (scrRNA-seq) the authors suggest that the activated HFSCs are under proliferation stress during hair regeneration. In vivo genetic studies further demonstrate that the loss of a single Bak allele (a pro-apoptotic protein) entirely rescues Mcl-1 deletion-associated defects in both HFs and mammary glands, indicating that these phenotypes are due to its anti-apoptotic function. Additionally, utilizing a Kinase screen and afatinib, a tyrosine kinase inhibitor of ERBB signaling, that authors found that ERBB pathway acts as an upstream regulator and plays a central function in promoting the survival of activated HFSCs and adult hair regeneration through promoting MCL-1 protein expression (Translational control and not transcriptional). The results are noteworthy, and the methods are sound. The presented work will be of significance in the field of skin and stem cell biology, HF morphogenesis and cycling.

Several questions remain to be addressed as listed below before the article could be considered for publication:

1. The inability of MCL-1 to not impact and alter the CD34+ HFSCs during quiescence is intriguing. Data suggest that K5-Cre mediated ablation of MCL-1 primarily affects and exhausts the "activated HFSCs" by a mechanism mediated by apoptosis. Although ligand inducible acute loss of MCL-1 abrogates the entire CD34+ /Sca-1- HFSC population. This dichotomy needs to be addressed and explained at a molecular level.
2. The authors acknowledge the presence of different subpopulation of stem cells and elegantly demonstrates expression of different protein markers using multiplex-IHC. However, the expression pattern of key HFSC markers such as NFATc1, LRIG1, LHX2, LGR5 and SOX9 in the upper and lower bulge region is not shown during the different stages of hair cycling in absence MCL-1. This could shed some light on the selective role of MCL-1 on the activated HFSC stem cells.
3. Stem cell activation into anagen is a tightly regulated event emerging from several positive and negative signals. BMP, NFAT, and FGF18 signaling provide strong inhibitory signals that keep the bulge and HG quiescent, whereas Wnt and FGF7 provide activation signals that overcome quiescence signals. Have the authors tested their expression in the HFs following constitutive and/or acute ablation of MCL-1?
4. Although MCL-1 is highly expressed in the interfollicular epidermis the phenotypic analysis is primarily based on HF morphogenesis, cycling and hair regeneration. Have the authors tested the function of MCL-1 on barrier integrity? TEWL analysis in the absence of MCL-1 protein and after abrasion of the epidermal barrier following tape stripping or chemical (acetone) treatment need to be performed. It is possible that barrier reestablishment could be impaired in absence of MCL-1 protein. What about any structural defects at the level of desmosomes, hemidesmosome or corneodesmosomes?
5. The impact of MCL-1 ablation (both constitutive and inducible) on the slow cycling BrdU+ label retaining cell (LRC) population need to be shown. Along the same line, it would be important to determine if there is any change in the LRC population following the rescue of the Mcl-1 deletion-associated defects after loss of a single Bak allele.
6. Why BAK was selected to study its role as the dominant downstream pro-apoptotic effector during HF morphogenesis and regeneration when it is already known that both BAK and BAX protein is inhibited by MCL-1? Especially, MCL-1 can physically bind to and inhibit the activity of BAX. Could there be a discrepancy between results from prior in vitro studies and in vivo phenotype observed with loss of BAX and BAK protein? Was there no effects on rescue of the Mcl-1 deletion phenotype following loss of Bax? The authors demonstrate that there is an increase in expression of both BAX and BAK protein in the CD49f+ |Sca-1+ interfollicular cell population in the absence of MCL-1.
7. The authors present data of the RNA velocity plot of the integrated HF cells of all samples during hair regeneration. How does Mcl-1 deletion impact the RNA velocity of the genes implicated in HF morphogenesis and hair cycling? Does that alteration reverse following loss of a Bak allele?

8. The description of the figures 6D and 6 E might have been reversed in the legend (minor). A careful check can resolve this discrepancy.
 9. There is little to no mention of IHC for P-Cad and use of DAPI as a counterstain of cell nuclei for fluorescent imaging in the legend for the main and supplementary figures. (minor)
 10. The EGFR inhibitor screen did not include Afatinib, which was later used for the inhibition of MCL-1 and the follow up pharmacological inhibition of hair cycling and stem cell activation. The rationale for this choice is not clear.
 11. The description of the color code representing the different genotype for the representative FACS plot is missing for figure S3 panel G (minor).
 12. The description of the legend for Figure S6 panel C, D and E don't match with the actual figure panels (C-E). This needs to be corrected and all figure panels for Figure S6 need to be labeled and described correctly in the legend to avoid confusion.
- shown during the different stages of hair cycling in absence MCL-1. This could shed some light on the selective role of MCL-1 on the activated HFSC stem cells.
3. Stem cell activation into anagen is a tightly regulated event emerging from several positive and negative signals. BMP, NFAT, and FGF18 signaling provide strong inhibitory signals that keep the bulge and HG quiescent, whereas Wnt and FGF7 provide activation signals that overcome quiescence signals. Have the authors tested their expression in the HFSCs following constitutive and/or acute ablation of MCL-1?
 4. Although MCL-1 is highly expressed in the interfollicular epidermis the phenotypic analysis is primarily based on HF morphogenesis, cycling and hair regeneration. Have the authors tested the function of MCL-1 on barrier integrity? TEWL analysis in the absence of MCL-1 protein and after abrasion of the epidermal barrier following tape stripping or chemical (acetone) treatment need to be performed. It is possible that barrier reestablishment could be impaired in absence of MCL-1 protein. What about any structural defects at the level of desmosomes, hemidesmosomes or corneodesmosomes?
 5. The impact of MCL-1 ablation (both constitutive and inducible) on the slow cycling BrdU+ label retaining cell (LRC) population needs to be shown. Along the same line, it would be important to determine if there is any change in the LRC population following the rescue of the Mcl-1 deletion-associated defects after loss of a single Bak allele.
 6. Why BAK was selected to study its role as the dominant downstream pro-apoptotic effector during HF morphogenesis and regeneration when it is already known that both BAK and BAX protein is inhibited by MCL-1? Especially, MCL-1 can physically bind to and inhibit the activity of BAX. Could there be a discrepancy between results from prior in vitro studies and in vivo phenotype observed with loss of BAX and BAK protein? Was there no effect on rescue of the Mcl-1 deletion phenotype following loss of Bax? The authors demonstrate that there is an increase in expression of both BAX and BAK protein in the CD49f+ |Sca-1+ interfollicular cell population in the absence of MCL-1.
 7. The authors present data of the RNA velocity plot of the integrated HF cells of all samples during hair regeneration. How does Mcl-1 deletion impact the RNA velocity of the genes implicated in HF morphogenesis and hair cycling? Does that alteration reverse following loss of a Bak allele?
 8. The description of the figures 6D and 6 E might have been reversed in the legend (minor). A careful check can resolve this discrepancy.
 9. There is little to no mention of IHC for P-Cad and use of DAPI as a counterstain of cell nuclei for fluorescent imaging in the legend for the main and supplementary figures. (minor)
 10. The EGFR inhibitor screen did not include Afatinib, which was later used for the inhibition of MCL-1 and the follow up pharmacological inhibition of hair cycling and stem cell activation. The rationale for this choice is not clear.
 11. The description of the color code representing the different genotype for the representative FACS plot is missing for figure S3 panel G (minor).
 12. The description of the legend for Figure S6 panel C, D and E don't match with the actual figure panels (C-E). This needs to be corrected and all figure panels for Figure S6 need to be labeled and described correctly in the legend to avoid confusion.

Reviewer #3

(Remarks to the Author)

In the manuscript by Chin et al., the authors explored the function of an anti-apoptotic protein, MCL-1, in the development and regeneration of hair follicles in mice. Using a combination of genetic mouse models, immunofluorescence staining, and single-cell RNAseq, the authors discovered that loss of MCL-1 leads to the loss of hair follicle stem cells (HFSCs), causing hair loss. Furthermore, the loss of HFSCs is specific to activated stem cells, as the quiescent stem cells remain intact after the acute ablation of MCL-1. Further investigation of the reason behind the stem cell loss discovered transcriptional changes associated with the loss of MCL-1 in HFSCs and the upstream ERBB pathway in regulating MCL-1 in HFSCs. Manipulating this pathway can mimic the effect of MCL-1 knockout, and the loss of the Bak gene together can rescue the defects associated with the loss of MCL-1. Overall, the experiments were well-designed and performed. Major concerns are related to the interpretation of the results and how these findings advance our current understanding of the field.

Suggestions:

1. The author comprehensively analyzed their single-cell data; however, the major reasons for the apoptosis of HFSCs are unclear. In addition, how is this mechanism related to the loss of MCL-1 on the molecular mechanism level? The authors should use both experiments and discussion to clarify these points.
2. The definition of "proliferation stress" in this context is unclear. Also, is there a mechanism for how MCL-1 can counteract this?

3. The authors used K5-Cre and K5-CreER to deplete MCL-1. Do these Cre lines specifically target the HFSC compartment? In other words, is the loss of HFSCs caused by the loss of MCL-1 within HFSCs or their niches?
4. The apoptotic HFSCs during anagen are quite interesting. How does this phenomenon lead to the loss of the hair follicle structure? Can the authors demonstrate the dynamic of hair follicles that eventually leads to losing the HF structure?
5. Does the K5-Cre also target the epidermis? If yes, do they also suffer a loss of epidermal stem cells since the epidermis is under constant self-renewal driven by activated epidermal stem cells?
6. Figure 1C KRT14 should be green in color.

Version 1:

Reviewer comments:

Reviewer #1

(Remarks to the Author)

Chin et al. have addressed a number of concerns and made meaningful improvements during the review process. In particular, the addition of the Mcl-1/Trp53 double knockout model in Figure 6 showing Trp53 loss can functionally rescue hair defects resulting from Mcl-1 deletion strengthens this manuscript. If possible, these results should be integrated together with Bak rescue experiments to provide a more wholistic view of intrinsic apoptosis (and its suppression via MCL-1) during stem cell-mediated hair growth. Efforts to address other major concerns (e.g. hair phenotypes are secondary to barrier defects, more targeted scRNA-sequencing analyses, and ERBB-targeting small molecule specificities) are satisfactory.

This said, there is still a lingering concern about the absence of a phenotype using the Lgr5-CreER model to delete Mcl-1. While it is possible that low recombination efficiencies are at fault, only two days of tamoxifen prior to waxing may be insufficient to achieve maximal efficiency. Additionally, if crossed with R26-TdTomato, do the authors observe apoptosis specifically in lineage-traced cells? Or their selective depletion over time after initiating hair cycling? If possible, either or both experiments should be performed and importantly, all data with this model included in the manuscript.

On more minor notes:

The authors should re-word lines 342-344 ("Confocal microscopy further showed that p-ERBB2 expression in hair follicles was high during anagen, with a pattern overlapping MCL-1, while it decreased in catagen and was low in telogen as previously reported, paralleling MCL-1 dynamics (Figure S9A)."). It is misleading as the ERBB2 expression referenced is not in the K14+ outer root sheath or HFSC compartment, where MCL-1 is in the anagen hair follicle. As it is, this data is not actually supportive of the hypothesis that p-ERBB2 is upstream of MCL-1 expression, and suggests that one of the other ERBB-family members may be key (eg EGFR). Perhaps re-framing Figure S9A in this light would be more helpful.

In line 486 of the discussion, there is a typo: ""mutant strains of with EGFR signaling".

Reviewer #2

(Remarks to the Author)

The current revised article by the group of Dr. Naiyang Fu and collaborators uncovers an essential role of the pro-survival protein MCL-1 in regulating HF development and adult hair regeneration. PI and collaborators utilize mouse genetics approach to demonstrate that MCL-1 is critical for the survival and function of activated HF stem cells (HFSCs) during hair regeneration. They demonstrate that constitutive deletion of Mcl-1 resulted in gradual hair loss and the eventual elimination of HFSCs.

Additionally, utilizing a Kinase screen, and two tyrosine kinase inhibitor of ERBB signaling (afatinib and erlotinib), that authors found that ERBB pathway acts as an upstream regulator and plays a central function in promoting the survival of activated HFSCs and adult hair regeneration through promoting MCL-1 protein expression (Translational control and not transcriptional).

The results are noteworthy, and the methods are sound. The presented work will be of significance in the field of skin and stem cell biology, HF morphogenesis and cycling.

The authors have successfully addressed all the concerns from the earlier submission. No additional concerns remain.

The results are noteworthy and the presented work will be of significance to the field of stem cell biology and related fields of HF biogenesis and regeneration.

Reviewer #3

(Remarks to the Author)

The authors have addressed all my questions, and I have no further inquiries regarding the manuscript. I recommend its

acceptance.

Version 2:

Reviewer comments:

Reviewer #1

(Remarks to the Author)

The authors have adequately addressed the remaining lingering concerns.

Reviewer #1 (Remarks to the Author):

In this manuscript, Chin et al. demonstrate that the anti-apoptotic protein MCL1 is required for hair follicle stem cell (HFSC) survival. Conditional loss of *Mcl1* from skin epithelium impaired acute hair regeneration and led to progressive hair loss, depleting HFSCs once activated in both contexts. A kinase screen identified ERBB signaling as an upstream regulator of MCL1 in keratinocytes with the potential to affect hair regeneration. Downstream, parallel deletion of BCL2 family members Bim and Bak partially and fully rescued HFSC number and hair regeneration defects, respectively.

In general, the manuscript is well executed, with reasonably comprehensive phenotyping and pharmacological and genetic phenocopying and rescue experiments lending credence to the ERBB - MCL1-Bim-Bak pathway mediating survival decisions of HFSCs. However, some major concerns should be addressed to bolster the manuscript (below).

Major concerns:

1) Cre-mediated conditional deletion was achieved using a promoter expressed in all skin epithelium, leaving open the possibility that any observed hair defects are secondary. In the absence of a HFSC-specific driver, the authors should be sure that *Mcl1* loss does not result in barrier breach, immune invasion or any other abnormality that can be assigned to the interfollicular epidermis. Is any effect (e.g. proliferation, stratification, etc.) observed in epidermal stem cells (EpdSCs)?

We thank the reviewer for the insightful comment regarding the possibility of secondary effects resulting from CRE-mediated conditional deletion using a promoter expressed throughout the skin epithelium. We acknowledge that it is important to confirm that the observed hair defects in *Mcl1*-deficient mice are specific to HFSCs and not indirect consequences of broader epidermal perturbations, such as barrier disruption, immune infiltration, or abnormalities in the interfollicular epidermis (IFE).

To address this concern, we collaborated with Dr. Prabha Sampath's team at A*STAR Skin Research Labs, experts in skin barrier function, inflammation and repair. We performed TransEpidermal Water Loss (TEWL) assays on our *K5-Cre^{ER}* inducible *Mcl1* knockout mice, and the results revealed no difference between *Mcl1* knockout and control littermates during the period of hair follicle (HF) regeneration following depilation (Figure S5A). Additionally, *Mcl1* deletion did not impact the KRT14⁺ basal epithelium or the KRT10⁺ terminally differentiated epidermis. Proliferation within the basal KRT14⁺ IFE was also comparable between knockout and control mice (Figure S5B&D). As shown in Figure 4C & Figure S5F, neither H&E staining nor immunostaining for immune markers indicated inflammation or immune cell infiltration. Furthermore, no significant apoptosis was detected in the IFE compared to HF (Figure S4A). These findings together clearly indicate that *Mcl1* deletion does not cause phenotypes associated with barrier disruption, immune infiltration or other abnormalities in the IFE.

As pointed out by the reviewer, basal cell markers, such as KRT5 and KRT14, are constitutively expressed in the IFE throughout all stages of HF development. However, upon activation, HFSCs rapidly lose the expression of these basal cell markers as they differentiate into transit-amplifying cells (TACs), which fuel subsequent HF regeneration and hair growth (Hsu *et al.*, *Cell*, 2014; PMID 24813615). We leveraged this distinct characteristic of HFSC differentiation and induced acute deletion of *Mcl-1* specifically after HFSC activation and commitment in our *K5-Cre^{ER}/Mcl-1* model. By administering tamoxifen at 3 days post depilation (dpd), a time point when HFSCs have transitioned to TACs, we ensured that *Mcl-1* was still efficiently deleted in the IFE, but MCL-1 expression remained intact in the majority of HF epithelial cells, including TACs, due to the absence of KRT5 expression in these more differentiated cells during anagen. Remarkably, under this condition, HF regeneration was indistinguishable between *Mcl-1* knockout and littermate control mice (Figure S4G-K). This demonstrates that HF regeneration is only significantly impaired when *Mcl-1* is deleted before HFSCs commit to more differentiated lineages. Given that *Mcl-1* deletion did not display phenotypes associated with dysregulation of epidermal stem cells (EpdSCs) or IFE in this experimental setup, these results further suggest that the observed HF defects are specifically due to *Mcl-1* loss in HFSCs and not due to secondary effects from the IFE.

The HFSC compartment is heterogenous, and it is known in the field that there is currently a lack of powerful inducible *Cre* systems that can delete genes both efficiently and specifically in all HFSC subpopulations—efficiency and specificity are both crucial. Given the well-recognized heterogeneity of HFSCs in the bulge region, *Krt5-/-Krt14-Cre* mouse models remain widely employed to investigate HFSC biology (Hsu *et al.*, *Nat Med*, 2014; PMID 25100530) (Hsu *et al.*, *Cell*, 2011; PMID 21215372) (Kretzschmar & Watt, *Cell*, 2012; PMID 22265400). Nevertheless, to explore the possibility of using a more HFSC-specific driver, we generated a conditional knockout model, *Lgr5-Cre^{ER}/Mcl-1^{fl/fl}*, to specifically assess the role of MCL-1 in the *Lgr5*⁺ cycling HFSC subpopulation (Jaks *et al.*, *Nat Genet*, 2008; PMID 18849992) (Hoeck *et al.*, *Nat Cell Biol*, 2017; PMID 28553937). Following *Mcl-1* deletion in *Lgr5*⁺ HFSCs, hair regeneration after depilation remained unaffected (Rebuttal Figure 1). Notably, *Mcl-1* deletion only occurred in a proportion of HF cells in this model. We further crossed *Lgr5-Cre* mice with *Rosa26-tdTomato* mice to examine the deletion efficiency in different subset of the skin epithelial cells by the *Lgr5-Cre^{ER}* system. TOMATO expression was found in ~50% and ~10% of the HFSCs (*Lin*⁻/*CD49f*⁺/*CD34*⁺/*SCA-1*⁻) and the non-defined HF (*Lin*⁻/*CD49f*⁺/*CD34*⁻/*SCA-1*⁻) populations respectively, but not in the IFE cells (*Lin*⁻/*CD49f*⁺/*SCA-1*⁺). Confocal analysis further confirmed that TOMATO expression was only switched on in a small proportion of HF cells. Collectively, these results suggest either a functional redundancy within distinct HFSC populations or a redundancy of distinct pro-survival Bcl-2 family members for the survival of *Lgr5*⁺ HFSCs and their progeny. We prefer not to include this data in our manuscript as we believe more comprehensive studies with inducible *Cre* system driven by the promoter of different HFSC marker genes are required to precisely map out the HFSC subpopulation(s) that are dependent on MCL-1 for their survival and contribute to the observed HF defects in this study, a process that would require significant resources and budget, and could take a few years to fully resolve.

Rebuttal Figure 1: Loss of *Mcl-1* in the *Lgr5*⁺ HFSC subpopulation does not result in HF regeneration defect

(A) *Lgr5-Cre^{ER}/Mcl-1^{fl/fl}* mice were administered with tamoxifen containing diet from p55 throughout the course of the experiment. A portion of the dorsal skin was shaved and waxed to induce hair regeneration. Skin samples were analysed 15 dpd. (B, C) Comparable hair regeneration between *Lgr5-Cre^{ER}/Mcl-1^{fl/fl}* mice and *Mcl-1^{fl/fl}* as shown by H&E histological sections (B), and confocal imaging (C). Representative confocal images show that a significant proportion of HF cells of *Lgr5-Cre^{ER}/Mcl-1^{fl/fl}* mice maintained the expression of MCL-1 (Left panel), even though *Lgr5-Cre^{ER}* mediated recombination as indicated by hCD4 expression was observed (Right panel). n = 6-7 mice per genotype. (D) Representative confocal images showing that the KRT15⁺ HFSC subpopulation is still present in *Lgr5-Cre/Mcl-1^{fl/fl}* mice. (E, F) Only a proportion of CD34⁺ HFSCs (CD34⁺/SCA-1⁻), and a small

proportion of non-defined HF cells (CD34⁺/SCA-1⁻) expresses TOMATO (RFP) in the *Lgr5-Cre^{ER}/tdTomato^{KI/+}* mice, as analysed by FACS (E) and confocal image (F).

2) It is unclear why activated HFSCs would require MCL 1 mediated protection. The authors reference the “stress” associated with stem cell activation, but little is done empirically to show how might interface with these pathways. Do proliferating HFSCs experience DNA damage, or any of the stress pathways identified in the scRNAseq? Any work to this end could offer real insight into the contextual regulators of apoptotic processes, significantly improving this manuscript.

We thank the reviewer for raising this important point regarding the requirement for MCL-1-mediated protection in activated HFSCs. In the previous version of our manuscript, the scRNA-seq analysis revealed that key pathways associated with cell cycle regulation, DNA replication and mismatch repair, cellular senescence, and P53 signaling are significantly enriched in the OB1 cluster consisting of CD34⁺ HFSC. To highlight the enrichment of these pathways and genes in activated HFSCs, we have now showed a heat map in the revised manuscript (Figure 5F). To provide direct evidence supporting these findings, we further performed γ H2AX (p-H2A.X) staining, a sensitive marker for DNA damage response, in HFs during anagen post depilation and confirmed that DNA damage response is indeed triggered in proliferating HFSCs (Figure 6A&B).

P53 is known to be a key regulator of proliferation stress and the DNA damage response. We further investigated its role in regulating HFSC function and survival in the context of *Mcl-1* deficiency. We have generated a new *K5-Cre^{ER}/Mcl-1^{fl/fl}/Trp53^{fl/fl}* inducible double knockout mouse model. Remarkably, we found that *Trp53* deletion rescued the HF regeneration defects caused by *Mcl-1* deficiency upon depilation in adult mice. New data from this model have now been included in the revised manuscript (Figure 6C-H). These new findings provide a more comprehensive understanding of the apoptotic regulatory processes during early hair regeneration in adult mice. The interplay between P53 and MCL-1 offers novel molecular insights into how cell proliferation and death are finely balanced in stem cells upon activation, which significantly advances the current understanding of stem cell biology in various tissues during regeneration. Interestingly, it has been previously shown that P53-mediated DNA damage response is triggered when quiescent lung stem cells are activated and undergo a transient state for regeneration while it remains to be determined whether MCL-1 may play an important role in the process (Kobayashi *et al.*, *Nat Cell Biol*, 2020; PMID 32661339).

3) Analyses of scRNA-sequencing from all cell types derived from depilation-induced skin do not fit well with the rest of the manuscript. The comparisons made between cell types and across time points do not yield information relevant to the author’s major conclusions or state somewhat obvious points, such as the upregulation of cell cycle-related genes following stem cell activation. It may be better utilized to identify potential HFSCs stressors which would necessitate Mcl1 anti-apoptotic activity during proliferation. Finally, the data set’s potential utility in identifying local sources of ERBB ligands or to dissect the dynamics of key pathway components during stem cell activation are underdeveloped and should be improved.

We thank the reviewer for this valuable comment. We agree that the connection between the scRNA-seq data and the rest of the manuscript was not made clear in the previous version of our manuscript. While scRNA-seq provides insights across all cell types in depilation-induced skin, our data reveal that the OB1 cell cluster—comprising quiescent stem cells at telogen and TACs at early anagen—is the most dynamic population. OB1 cells are the primary drivers of adult hair regeneration (Figure 5E). Our analysis showed that, upon activation, upregulated genes in OB1 are associated with cell cycle, DNA replication, DNA repair, cellular senescence, and the P53 signaling pathway (Figure S7D, Left). This scRNA-seq data motivated us to generate the new *K5Cre^{ER}/Mcl-1^{ff}/Trp53^{ff}* inducible double knockout model, which revealed that P53 acts as a key sensor of proliferation stress in HFSCs, while MCL-1 is essential for balancing proliferation and cell death during regeneration (Figure 6C-H).

We also explored cell-cell communication in our scRNA-seq data to identify potential sources of ERBB ligands and other signaling pathways using multiple pipelines. However, we believe more detailed experiments, such as spatial omics, are necessary for conclusive results. Including these preliminary or inconclusive data could detract from the manuscript's focus. Most importantly, we have now strengthened the link between the identification of DNA mismatch and P53 signaling by our scRNA-seq data and the manuscript's main findings. Once our paper is published, our scRNA-seq dataset will be publicly available, representing the first comprehensive dataset for adult HF regeneration upon depilation, which we hope would provide a valuable resource for the field and stimulate further discoveries from our data by using various analytical tools.

4) The use of afatinib, a tyrosine kinase inhibitor with relative HER2/4 specificity, is relied on too heavily to substantiate claims that ERBB regulates MCL1 expression and blocks hair regeneration. The finding that afatinib broadly impacts the activation of protein translational machinery, in particular, creates some doubt that these effects are solely the result of ERBB inhibition. These data should be supplemented with a more thorough, unbiased examination of its potential off-target effects and more specific (e.g. genetic) evidence, at least in cultured keratinocytes, that ERBB signaling is responsible for regulating MCL1 expression. Rescue experiments are needed to demonstrate that any effects on keratinocyte apoptosis are specific to MCL1.

We thank the reviewer for the thorough and constructive feedback. We recognize the concern regarding our reliance on afatinib to support the role of ERBB in regulating MCL-1 expression and its impact on hair regeneration. We would like to clarify that one of the primary outcomes of ERBB pathway activation is the stimulation of the mTOR pathway, which selectively drives the translation of specific mRNAs, including *Mcl-1* (Mills *et al.*, *PNAS*, 2008; PMID 18664580) (Winder & Campbell, *Cell Cycle*, 2022; PMID 35349392). Therefore, inhibition of mTOR and translation by afatinib should not be considered an off-target effect but rather a downstream consequence of ERBB inhibition. Nevertheless, we have incorporated an additional ERBB-specific inhibitor, erlotinib, into our analysis to address these concerns. Erlotinib, which was also identified in our kinase inhibitor screen, significantly reduced MCL-1 protein levels and impaired hair regeneration post depilation. These results have been added to the revised manuscript (Figure S9F&G). Aside from the studies presented here, compelling evidence highlights the crucial role of ERBB signaling in regulating MCL-1 translation in various

tissues. For example, we previously reported that ERBB inhibition downregulates MCL-1 protein levels in lactating mammary gland (Fu *et al.*, *Nat Cell Biol*, 2015; PMID 25730472).

We appreciate the reviewer's suggestion to supplement our study with a genetic approach in cultured keratinocytes. However, this has proved experimentally difficult as primary murine keratinocytes are challenging to maintain beyond two weeks due to spontaneous cell death and are reluctant to transfection and infection. However, as discussed in our manuscript, previous genetic studies have provided strong evidence for the *in vivo* role of ERBB signaling in hair regeneration. For example, overexpression of a dominant-negative EGFR mutant in KRT5⁺ cells did not perturb HF morphogenesis but resulted in progressive hair loss (Murillas *et al.*, *EMBO*, 1995; PMID 7489711). Additionally, EGF overexpression disrupts the transition from anagen to catagen, a key step in the hair cycle, leading to continuous hair follicle growth without the normal regression phase (Mak & Chan, *J Biological Chemistry*, 2003; PMID 12714603). Generating a new *Mcl-1* overexpression mouse line would be the best way for *in vivo* rescue experiments, but it would take at least another 1-2 years. More importantly, we would like to emphasize that we do not suggest in our manuscript that MCL-1 is the sole mediator of ERBB signaling in hair regeneration. Rather, our data highlight that ERBB signaling plays a crucial role in HFSC maintenance and hair regeneration, at least in part by regulating MCL-1 protein translation.

Minor points:

A) In figure 1 the authors refer to “early” anagen which is conventionally when HFSCs proliferate (anagen II-III); however, the representative images shown are mid-late anagen (at least anagen IV). Data from earlier anagen should be included for *Mcl1*, *Bclxl*, *Bak*, *Bim*. For accurate staging please see: Müller-Röver, S. et al. A Comprehensive Guide for the Accurate Classification of Murine Hair Follicles in Distinct Hair Cycle Stages. *Journal of Investigative Dermatology* 117, 3-15 (2001).

We thank the reviewer for this valuable suggestion. In response, we have now included an early anagen (Anagen II-III) timepoint in Figure 1, following the classification guidelines from Müller-Röver et al. (*Journal of Investigative Dermatology*, 2001). Additionally, we have updated the other panels to accurately reflect the hair follicle staging in Figure 1.

B) Do the authors have any insight into why *Mcl1* loss has no effect during development? Or why only the HFSC population is affected and not EpdSCs (assuming this is the casual phenotype)? Elaboration in the discussion would be useful.

We thank the reviewer for this insightful comment. CD34⁺ quiescent HFSCs only emerge after HF organogenesis and play an essential role in subsequent hair follicle cycles in adult mice (Trempeus *et al.*, *J Invest Dermatol*, 2003; PMID 12648211). Recent studies indicate that while HFSCs contribute to the renewal and maintenance of hair follicles, the initial stages of HF organogenesis are primarily influenced by neuroectodermal-mesodermal interactions and the surrounding microenvironment, rather than by these stem

cells. This is supported by findings that highlight the importance of mesenchymal condensates and gradients of signaling molecules, such as WNT, SHH, and BMP, in directing HF development (Schmidt-Ullrich & Paus, *Bioessays*, 2005; PMID 15714560) (Nowak & Fuchs, *Methods Mol Biol*, 2009; PMID 19089359) (Schneider *et al.*, *Curr Biol*, 2009; PMID 19211055). We believe that *Mcl-1* loss has no effect during HF organogenesis or neonatal HF development because these processes are not driven by HFSCs. We agree with the reviewer that it is an interesting topic to address why MCL-1 is dispensable during early development. But this would require extensive future investigation, which falls outside the scope of our current focus on the role of MCL-1 in adult HF regeneration.

As noted in response to your Question 1, deletion of *Mcl-1* in the IFE does not impact IFE cell survival, skin integrity, or cycling capacity, suggesting that different cell types in the epidermis have distinct pro-survival mechanisms. It is well established that even within the same tissue, different cell types rely on different pro-survival proteins. For instance, during T cell development, thymocytes at different stages rely on different BCL-2 family members: MCL-1 is essential for DN2 thymocytes, BCL-XL for DP T cells, and BCL-2 for mature, unstimulated T cells (Nakayama *et al.*, *Science*, 1993; PMID 8372353) (Dzhagalov *et al.*, *J Immunol*, 2008; PMID 18566418) (Tripathi *et al.*, *Cell Death Differ*, 2013; PMID 23558951). Interestingly, the upregulation of a pro-survival protein does not necessarily indicate its requirement for cell survival. For example, although A1 is upregulated upon T cell receptor engagement, its genetic deletion does not affect T cell-driven immune responses (Vershelde *et al.*, *Cell Death & Differentiation*, 2003; PMID 12934080) (Tuzlak *et al.*, *Cell Death & Differentiation*, 2017; PMID 28085151) (Schenk *et al.*, *Cell Death Differ*, 2017; PMID 28085150). These examples illustrate the complex and dynamic regulation of pro-survival proteins across different cell types and developmental stages. We have now elaborated on these points in the Discussion section of the revised manuscript, as per the reviewer's suggestion.

C) Quantification of phenotypes graphically or definitive statement of percentage of phenotypic animals (eg with greater than 50% hair loss) under each condition should be added in the text.

Please note that no significant variation was observed in the phenotypes between mice with the same genotype under each condition. The photos of mice provide visible, but not quantitative phenotypes. The phenotypes between control and knockout mice are so profound. We hence did not conduct quantitative analysis on hair loss. We stated in the text that mice are hairless when it is appropriate.

D) Figure 3A,B is confusing to interpret. Why not present percentage of each cell type (HFSC, IFE, other) of the total, instead of % Cd49f against the other 2 markers?

We thank the reviewer for pointing this out, as such we have amended “% CD49f⁺” to “% in CD49f⁺ cells” to make it clearer. We have amended this in Figures 3B, 4H, 4J, 7H, 9H, S3F, S9C.

E) Typo in line 107 (“depletio”)

We have rectified this mistake.

F) Data regarding loss of Cd34+ HFSCs, especially with respect to the depilation experiment, needs some clarification. Is it truly loss of the HFSCs in the *Mcl1* conditional knockout within 3 days post depilation, or loss of the CD34 marker? Additional markers of HFSC identity should be investigated (K24, *Nfatc1*, *Sox9*, etc.)

While our studies mainly utilized CD34 as a marker for bulge stem cells, we have now included KRT15 (Liu *et al.*, *J Invest Dermatol*, 2003; PMID 14708593), SOX9 (Vidal *et al.*, *Curr Biol*, 2005; PMID 16085486) and LHX2 (Rhee *et al.*, *Science*, 2006; PMID 16809539) as markers of bulge stem cells to determine if these are present in 9 dpd skin when *Mcl-1* is deleted. In parallel with our initial results, *Mcl-1* deletion results in a dramatic loss of KRT15, SOX9, and LHX2 bulge stem cells, respectively, in HF 9 dpd (Figure S4C-E). Hence, providing compelling evidence that *Mcl-1* deletion results in the complete loss of HFSC post depilation rather than the loss of the stem cell marker CD34.

We acknowledge that NFATc1 is also a recognized marker of HFSCs. However, despite attempting to stain for NFATc1 using antibodies from Santa Cruz (mouse, 7A6, 1:100) and Proteintech (mouse, 1E1B10, 1:150), we were unable to detect any signal. This may be due to limitations in paraffin embedment or batch variations in the antibodies.

G) Figure 6D and 6E references are switched in text (page 15-16).

We have now rectified this mistake.

Reviewer #2 (Remarks to the Author):

The current article by the group of Dr. Naiyang Fu and collaborators uncovers an essential role of the pro-survival protein MCL-1 in regulating HF development and adult hair regeneration. PI and collaborators utilize mouse genetics approach to demonstrate that MCL-1 is critical for the survival and function of activated HF stem cells (HFSCs) during hair regeneration. They demonstrate that constitutive deletion of *Mcl-1* resulted in gradual hair loss and the eventual elimination of HFSCs.

Results suggest that MCL-1 is dispensable for quiescent HFSCs, but its acute deletion during the activation phase results in rapid HFSC loss and a complete block in hair regrowth in adult mice. Using single-cell RNA-sequencing (scRNA-seq) the authors suggest that the activated HFSCs are under proliferation stress during hair regeneration. In vivo genetic studies further demonstrate that the loss of a single *Bak* allele (a pro-apoptotic protein) entirely rescues *Mcl-1* deletion-associated defects in both HFs and mammary glands, indicating that these phenotypes are due to its anti-apoptotic function.

Additionally, utilizing a Kinase screen and afatinib, a tyrosine kinase inhibitor of ERBB signaling, that authors found that ERBB pathway acts as an upstream regulator and plays a central function in promoting the survival of activated HFSCs and adult hair regeneration through promoting MCL-1 protein expression (Translational control and not transcriptional).

The results are noteworthy, and the methods are sound. The presented work will be of significance in the field of skin and stem cell biology, HF morphogenesis and cycling.

Several questions remain to be addressed as listed below before the article could be considered for publication:

1. The inability of MCL-1 to not impact and alter the CD34⁺ HFSCs during quiescence is intriguing. Data suggest that K5-Cre mediated ablation of MCL-1 primarily affects and exhaust the “activated HFSCs” by a mechanism mediated by apoptosis. Although ligand inducible acute loss of MCL-1 abrogates the entire CD34⁺/Sca-1- HFSC population. This dichotomy needs to be addressed and explained at a molecular level.

We thank the reviewer for raising this important point. As indicated, the impact of MCL-1 appears to be selective to activated HFSCs rather than quiescent CD34⁺ HFSCs. Without inducing regeneration, our data show that quiescent CD34⁺ HFSCs remain unaffected when *Mcl-1* is deleted, whereas activated HFSCs are rapidly depleted upon *Mcl-1* deletion following depilation.

During quiescence, HFSCs are largely dormant and not under the same proliferation-associated stress as activated HFSCs. Our scRNAseq data reveal that once activated, CD34⁺ HFSCs upregulate genes involved in cell cycle regulation, DNA replication, DNA repair, and the P53 signaling pathway (Figure 5F; Figure S7D, Left). This suggests that activated HFSCs experience significant proliferation stress, making them more dependent on anti-

apoptotic signals such as those provided by MCL-1. In contrast, quiescent HFSCs, which are not subjected to this stress, appear to tolerate the loss of *Mcl-1*, consistent with studies in other tissues where different survival proteins are required at various stages of cell activation and differentiation. For example, during T cell development, MCL-1 is critical for early DN2 thymocytes but dispensable for mature T cells, which rely on other pro-survival proteins such as BCL-XL and BCL-2 (Nakayama *et al.*, *Science*, 1993; PMID 8372353) (Dzhagalov *et al.*, *J Immunol*, 2008; PMID 18566418) (Tripathi *et al.*, *Cell Death Differ*, 2013; PMID 23558951).

The rapid depletion of CD34⁺ HFSCs cells upon acute deletion of *Mcl-1* in the depilation-induced hair regeneration model in adult mice is most likely because most of quiescent HFSCs are activated under the experimental condition. Our scRNA-seq analysis revealed that key pathways associated with cell cycle regulation, DNA replication and mismatch repair, cellular senescence, and P53 signaling are significantly enriched in the OB1 cluster consisting of CD34⁺ HFSC. To highlight the enrichment of these pathways and genes in activated HFSCs, we have now showed a heat map in the revised manuscript (Figure 5F). To provide direct evidence supporting these findings, we further performed γ H2AX (p-H2A.X) staining, a sensitive marker for DNA damage response, in HF during anagen post depilation and confirmed that DNA damage response is indeed triggered in proliferating HFSCs (Figure 6A&B). P53 is known to be a key regulator of proliferation stress and the DNA damage response. We further investigated its role in regulating HFSC function and survival in the context of *Mcl-1* deficiency. We have generated a new *K5-Cre^{ER}/Mcl-1^{ff}/Trp53^{ff}* inducible double knockout mouse model. Remarkably, we found that *Trp53* deletion rescued the HF regeneration defects caused by *Mcl-1* deficiency upon depilation in adult mice. New data from this model have now been included in the revised manuscript (Figure 6C-H). Moreover, these new findings from the *Mcl-1/Trp53* double knockout model indicate that MCL-1 plays a central role in protecting these cells from P53-dependent apoptosis under conditions of proliferation stress, highlighting the interplay between these two proteins in balancing cell proliferation and apoptosis. The inability of MCL-1 to affect quiescent HFSCs underscores that different stem cell states have distinct survival mechanisms, which are fine-tuned based on their activity level. But this dichotomy would require extensive future investigation.

2. The authors acknowledge the presence of different subpopulation of stem cells and elegantly demonstrates expression of different protein markers using multiplex-IHC. However, the expression pattern of key HFSC markers such as NFATc1, LRIG1, LHX2, LGR5 and SOX9 in the upper and lower bulge region is not shown during the different stages of hair cycling in absence MCL-1. This could shed some light on the selective role of MCL-1 on the activated HFSC stem cells.

We are grateful to the reviewer for this valuable comment. While our initial studies primarily utilized CD34 as a marker for bulge stem cells, we have now expanded our analysis to include additional HFSC markers including KRT15, SOX9 and LHX2 to assess their expression in the bulge region at 9 days post-depilation (dpd). Consistent with our initial findings, *Mcl-1* deletion resulted in a dramatic loss of KRT15, SOX9, and LHX2-positive bulge stem cells at all time points post-depilation (Figure S4C-E). This provides compelling

evidence that *Mcl-1* deletion leads to the complete depletion of HFSCs following depilation, underscoring its critical role in HFSC maintenance and survival during hair regeneration.

We acknowledge that NFATc1 is also a recognized marker of HFSCs. However, despite attempting to stain for NFATc1 using antibodies from Santa Cruz (mouse, 7A6, 1:100) and Proteintech (mouse, 1E1B10, 1:150), we were unable to detect any signal. This may be due to limitations in paraffin embedment or batch variations in the antibodies.

3. Stem cell activation into anagen is a tightly regulated event emerging from several positive and negative signals. BMP, NFAT, and FGF18 signaling provide strong inhibitory signals that keep the bulge and HG quiescent, whereas Wnt and FGF7 provide activation signals that overcome quiescence signals. Have the authors tested their expression in the HFs following constitutive and/or acute ablation of MCL-1?

We fully agree with the reviewer that adult HF regeneration is orchestrated by a complex interplay of positive and negative signals, including pathways such as BMP, NFAT, FGF18, Wnt, and FGF7, which regulate the balance between quiescence and activation of HFSCs. However, based on our findings, the primary defect in HF regeneration following *Mcl-1* ablation is linked to its critical pro-survival role at the mitochondria, as demonstrated by the complete rescue of HF regeneration defects in the *Bak* knockout model. This strongly suggests that the primary mechanism by which MCL-1 functions is through the regulation of apoptosis rather than direct modulation of these signaling pathways.

While we have not specifically tested changes in BMP, NFAT, FGF18, Wnt, or FGF7 signaling following *Mcl-1* ablation, our data indicate that the main driver of the observed HFSC depletion and impaired regeneration is the loss of HFSCs in the absence of *Mcl-1*. Given that *Bak* deletion fully rescues the hair regeneration phenotypes in the *Mcl-1*-deficient skin, it is unlikely that MCL-1 directly regulates these upstream pathways. Therefore, although further exploration into the signaling pathways mentioned would undoubtedly provide additional insights, we believe it would not significantly alter the key conclusion of our manuscript, which emphasizes the pivotal role of MCL-1 in preventing apoptosis during HF regeneration.

4. Although MCL-1 is highly expressed in the interfollicular epidermis the phenotypic analysis is primarily based on HF morphogenesis, cycling and hair regeneration. Have the authors tested the function of MCL-1 on barrier integrity? TEWL analysis in the absence of MCL-1 protein and after abrasion of the epidermal barrier following tape stripping or chemical (acetone) treatment need to be performed. It is possible that barrier reestablishment could be impaired in absence of MCL-1 protein. What about any structural defects at the level of desmosomes, hemidesmosome or corneodesmosomes?

We thank the reviewer for these valuable comments and acknowledge the importance of evaluating MCL-1 function in barrier integrity and potential structural defects in the skin. To address these concerns, we collaborated with Dr. Prabha Sampath's team at A*STAR Skin Research Labs, experts in skin barrier function, inflammation

and repair. We conducted TransEpidermal Water Loss (TEWL) analysis following shaving and waxing to depilate the dorsal skin of our K5Cre^{ER} inducible *Mcl-1* knockout model (Figure S5A). Our results indicate no significant differences in TEWL between *Mcl-1* knockout and control mice during HF regeneration following shaving and waxing. Additionally, *Mcl-1* deletion did not impact the KRT14⁺ basal epithelium or KRT10⁺ terminally differentiated epidermis (Figure S5B), and proliferation within the KRT14⁺ interfollicular epithelium (IFE) remained comparable between knockout and control mice. H&E staining and immunostaining for CD4 T cells, CD8 T cells and macrophages revealed no signs of inflammation or immune infiltration (Figure 4C; Figure S5C-F), and apoptosis was not significantly detected in the IFE (Figure S4A). Based on these results, *Mcl-1* deletion does not appear to compromise barrier function or skin structure under these conditions. Additionally, we did not observed differences in desmoplakin or plectin staining inferring that the desmosomes and hemidesmosomes were not impacted in the K5Cre^{ER} inducible *Mcl-1* knockout model (Figure S5C).

Basal cell markers, such as KRT5 and KRT14, are constitutively expressed in the IFE throughout all stages of HF development. However, upon activation, HFSCs rapidly lose the expression of these basal cell markers as they differentiate into transit-amplifying cells (TACs), which fuel subsequent HF regeneration and growth (Hsu & Fuchs, *Cell*, 2014; PMID 24813615). We leveraged this distinct characteristic of HFSC differentiation and induced acute deletion of *Mcl-1* specifically after HFSC activation and commitment in our *K5-Cre^{ER}* model. By administering tamoxifen 3 dpd, a time point when HFSCs have transitioned to TACs, we ensured that *Mcl-1* was still efficiently deleted in the IFE, but MCL-1 expression remained intact in the majority of epithelial cells, TACs, due to the absence of KRT5 expression in these more differentiated HFs during anagen. Remarkably, under these conditions, HF regeneration was indistinguishable between *Mcl-1* knockout and littermate control mice (Figure S4G-K). This indicates that HF regeneration is only disrupted when *Mcl-1* is deleted before HFSCs commit to differentiated lineages. Since *Mcl-1* deletion did not impact the interfollicular epidermis (IFE) in this experimental context, these findings provide strong evidence that the HF defects caused by Mcl-1 loss are not due to secondary effects in the IFE. Additionally, the data suggest that Mcl-1 deletion does not significantly affect the function or behavior of epidermal stem cells (EpdSCs), confirming that the hair regeneration defects are directly linked to the loss of *Mcl-1* in HFs.

5. The impact of MCL-1 ablation (both constitutive and inducible) on the slow cycling BrdU⁺ label retaining cell (LRC) population need to be shown. Along the same line, it would be important to determine if there is any change in the LRC population following the rescue of the Mcl-1 deletion-associated defects after loss of a single Bak allele.

We would like to highlight that BrdU⁺ label-retaining cells (LRCs) are commonly referred to as CD34⁺ quiescent HFSCs (Trempeus *et al.*, *J Invest Dermatol*, 2003; PMID 12648211) (Tumbar *et al.*, *Science*, 2004; PMID 14671312). As demonstrated in our study, CD34⁺ HFSCs are significantly reduced following *Mcl-1* ablation in both the constitutive and inducible knockout models. In addition to the depletion of CD34⁺ HFSCs, we observed a dramatic reduction in other key stem cell populations, including KRT15⁺, SOX9⁺, and LHX2⁺ HFSCs, during hair regeneration post-depilation in our *K5-Cre^{ER}/Mcl-1^{ff}* model. The rapid depletion HFSCs and subsequent

destruction of HF in the *K5-Cre^{ER}/Mcl-1^{ff}* knockout model precludes long-term studies on the label-retaining population in this context.

Moreover, our data show that the loss of a single *Bak* allele can fully rescue the defects associated with *Mcl-1* deficiency, including the restoration of the CD34⁺ HFSC population. This suggests that the primary mechanism underlying the loss of CD34⁺ HFSCs in *Mcl-1* deficient mice is driven by apoptosis mediated through the mitochondrial apoptotic pathway. This highlights the critical interplay between MCL-1 and pro-apoptotic proteins like BAK in maintaining the survival and function of HFSCs during hair regeneration.

6. Why BAK was selected to study its role as the dominant downstream pro-apoptotic effector during HF morphogenesis and regeneration when it is already known that both BAK and BAX protein is inhibited by MCL-1? Especially, MCL-1 can physically bind to and inhibit the activity of BAX. Could there be a discrepancy between results from prior in vitro studies and in vivo phenotype observed with loss of BAX and BAK protein? Was there no effects on rescue of the *Mcl-1* deletion phenotype following loss of *Bax*? The authors demonstrate that there is an increase in expression of both BAX and BAK protein in the CD49f⁺ |Sca-1⁺ interfollicular cell population in the absence of MCL-1.

We appreciate the reviewer's focus on the interplay between MCL-1 and BAK. More and more evidence has emerged to unmask the non-apoptotic functions of MCL-1. The gold standard to address whether a phenotype associated with *Mcl-1* deficiency is due to its anti-apoptotic function or not is to test whether knockout of proapoptotic BH3-only or BCL-2 pro-apoptotic effector proteins (i.e. BAK, BAX and BOK) is able to rescue the defect. Understanding the interplay between these pro-apoptotic proteins remains a crucial area of investigation within the BCL-2 family. In this study, our goal was to provide strong evidence that the defects in hair follicle regeneration caused by *Mcl-1* deficiency are due to its pro-survival function. Notably, knockout of a single *Bak* allele was sufficient to fully rescue the defects, strongly supporting the idea that these phenotypes arise due to the anti-apoptotic role of MCL-1.

We acknowledge the reviewer's point that MCL-1 can also inhibit BAX and potentially BOK, and this interplay could contribute to the observed phenotypes. While we did not focus on BAX or BOK in this study, we have included a discussion of their possible involvement in the revised manuscript. Whether MCL-1 predominantly restrains BAX, BAK, or BOK, or a combination of these proteins, remains an open question that will require further investigation to clarify the specific mechanisms by which MCL-1 regulates apoptosis in HFSCs.

7. The authors present data of the RNA velocity plot of the integrated HF cells of all samples during hair regeneration. How does *Mcl-1* deletion impact the RNA velocity of the genes implicated in HF morphogenesis and hair cycling? Does that alteration reverse following loss of a *Bak* allele?

We appreciate the reviewer's suggestions. CD34⁺ HFSCs sit at the apex of the stem cell differentiation hierarchy, and it is expected that their depletion would significantly disrupt this hierarchy. Our data clearly show

that CD34⁺ HFSCs are rapidly depleted, leading to the complete destruction of HF in the absence of *Mcl-1*. Given that the loss of a single *Bak* allele fully restores HF regeneration in *Mcl-1* deficient mice, we anticipate that the RNA velocity would revert to patterns similar to those observed in wild-type mice. Therefore, while conducting additional scRNA-seq experiments could confirm our hypothesis, we believe it is unlikely to yield any new insights beyond what is already evident from our current findings, especially considering the substantial costs and resources involved.

8. The description of the figures 6D and 6E might have been reversed in the legend (minor). A careful check can resolve this discrepancy.

We have now rectified this mistake.

9. There is little to no mention of IHC for P-Cad and use of DAPI as a counterstain of cell nuclei for fluorescent imaging in the legend for the main and supplementary figures. (minor)

We have now rectified this mistake.

10. The EGFR inhibitor screen did not include Afatinib, which was later use for the inhibition of MCL-1 and the follow up pharmacological inhibition of hair cycling and stem cell activation. The rationale for this choice is not clear.

We used afatinib in our study as it is an independent ERBB inhibitor from the screen and a clinically approved drug, offering translational relevance to our findings. However, to address the reviewer's concern, we have now included another ERBB inhibitor, erlotinib, in our analysis. Notably, erlotinib was part of the original kinase inhibitor screen used to identify compounds that downregulate MCL-1 protein expression in primary cultured keratinocytes. Consistent with afatinib, we found that erlotinib also significantly reduced MCL-1 protein levels and impaired adult hair regeneration following depilation. These results have been incorporated into the revised manuscript (Figure S9F&G).

11. The description of the color code representing the different genotype for the representative FACS plot is missing for figure S3 panel G (minor).

We have now rectified this mistake in the revised manuscript.

12. The description of the legend for Figure S6 panel C, D and E don't match with the actual figure panels (C-E). This need to be corrected and all figure panel for Figure S6 need to be labeled and described correctly in the legend to avoid confusion.

We have now rectified this mistake in the revised manuscript.

Reviewer #3 (Remarks to the Author):

In the manuscript by Chin et al., the authors explored the function of an anti-apoptotic protein, MCL-1, in the development and regeneration of hair follicles in mice. Using a combination of genetic mouse models, immunofluorescence staining, and single-cell RNAseq, the authors discovered that loss of MCL-1 leads to the loss of hair follicle stem cells (HFSCs), causing hair loss. Furthermore, the loss of HFSCs is specific to activated stem cells, as the quiescent stem cells remain intact after the acute ablation of MCL-1. Further investigation of the reason behind the stem cell loss discovered transcriptional changes associated with the loss of MCL-1 in HFSCs and the upstream ERBB pathway in regulating MCL-1 in HFSCs. Manipulating this pathway can mimic the effect of MCL-1 knockout, and the loss of the Bak gene together can rescue the defects associated with the loss of MCL-1. Overall, the experiments were well-designed and performed. Major concerns are related to the interpretation of the results and how these findings advance our current understanding of the field.

We appreciate the reviewer's thorough summary of our manuscript. In this study, we uncover the essential role of the pro-survival protein MCL-1 in regulating hair follicle (HF) development and adult hair regeneration. Using innovative genetic models, we demonstrate that MCL-1 is critical for the survival and function of activated hair follicle stem cells (HFSCs) during hair regeneration. Our single-cell RNA sequencing (scRNA-seq) analysis revealed that activated HFSCs experience proliferation stress during this process, with enrichment of p53 and DNA mismatch repair signaling pathways. We further elucidate the mechanism by showing that the loss of a single *Bak* allele, a pro-apoptotic protein, fully rescues the defects associated with *Mcl-1* deletion in both HFs and mammary glands, confirming that these phenotypes are linked to the anti-apoptotic function of MCL-1. Additionally, we highlight the pivotal role of the ERBB pathway in promoting the survival of activated HFSCs and adult hair regeneration by driving MCL-1 protein expression. Together, our findings emphasize the critical role of MCL-1 in preventing stress-induced apoptosis during HFSC activation, a process vital for adult hair regeneration. These insights significantly advance our understanding of the molecular mechanisms that govern HFSC function and adult tissue regeneration, with broader implications for stem cell biology and regenerative medicine.

Suggestions:

1. The author comprehensively analyzed their single-cell data; however, the major reasons for the apoptosis of HFSCs are unclear. In addition, how is this mechanism related to the loss of MCL-1 on the molecular mechanism level? The authors should use both experiments and discussion to clarify these points.

Thank you for raising this important point. Our scRNA-seq analysis revealed that key pathways associated with cell cycle regulation, DNA replication and mismatch repair, cellular senescence, and P53 signaling are significantly enriched when CD34⁺ HFSCs exit quiescence and re-enter cell cycle. This suggests that the augmented levels of apoptosis observed in *Mcl-1*-deficient HFSCs is likely driven by heightened proliferation stress and DNA damage response. To substantiate these findings, we further validated our scRNA-seq results by

performing γ H2AX (pH2A.X) staining on HF during anagen following depilation. This sensitive marker of DNA damage and repair was prominently detected in both activated HFSCs and transit-amplifying cells (TACs), providing molecular evidence of DNA damage response during the regenerative phase (Figure 6A&B; revised manuscript).

Given that P53 is a key sensor and regulator of proliferation stress and DNA damage responses, we explored the interplay between P53 and MCL-1 by generating a new inducible double knockout mouse model, *K5Cre^{ER}/Mcl-1^{ff}/Trp53^{ff}*. Remarkably, our data show that conditional deletion of *Trp53* rescues the hair regeneration defects caused by *Mcl-1* deficiency, highlighting the critical role of P53 in regulating apoptosis in response to stress during HFSC activation. These new findings clarify the molecular mechanism by which *Mcl-1* deficiency leads to HFSC apoptosis and emphasize the critical role of MCL-1 in mitigating P53-mediated apoptosis during the hair regeneration process and provide deeper insights into how MCL-1 ensures HFSC survival under proliferation stress, with broader implications for stem cell regulation in various tissues during regeneration.

2. The definition of “proliferation stress” in this context is unclear. Also, is there a mechanism for how MCL-1 can counteract this?

Proliferation stress refers to the cellular strain that occurs during rapid or sustained cell division, placing a high demand on the machinery responsible for DNA replication, repair, and genomic stability. Under these conditions, cells are more prone to replication errors, DNA damage, and chromosomal instability (Zeman & Cimprich, *Nat Cell Biol*, 2014; PMID 24366029). If not properly held in check, this can lead to apoptosis or senescence. This phenomenon is particularly significant in rapidly cycling cells, such as stem cells, cancer cells, and regenerating tissues, where the balance between proliferation and survival is crucial for maintaining tissue homeostasis (Simons & Clevers, *Cell*, 2011; PMID 21663791). The P53 signaling pathway serves as a central regulator in responding to proliferation stress by either activating DNA repair mechanisms or triggering apoptosis if the damage is irreparable (Bartek & Lukas, *Curr Opin Cell Biol*, 2007; PMID 17303408). To provide direct evidence supporting these findings, we further performed γ H2AX (p-H2A.X) staining, a sensitive marker for DNA damage response, in HFs during anagen post depilation and confirmed that DNA damage response is indeed triggered in proliferating HFSCs (Figure 6A&B). Moreover, our new data show that the conditional deletion of *Trp53* rescues the hair regeneration defects caused by *Mcl-1* deficiency, highlighting the role of P53 signaling in response to stem cell activation and stress. In this context, MCL-1, a pro-survival member of the BCL-2 family, plays a critical role in protecting cells from P53-mediated apoptosis during periods of proliferation stress, allowing HFSCs to survive during proliferation stress, which is critical for successful HF regeneration. The broader implications suggest that MCL-1 plays a pivotal role in regulating stem cell survival across various tissues under conditions of high proliferative demand, safeguarding against apoptosis and ensuring proper tissue maintenance and regeneration.

3. The authors used K5-Cre and K5-CreER to deplete MCL-1. Do these Cre lines specifically target the HFSC compartment? In other words, is the loss of HFSCs caused by the loss of MCL-1 within HFSCs or their niches?

We appreciate the reviewer's question regarding the specificity of the *K5-Cre* and *K5-Cre^{ER}* lines in targeting HFSCs. HFSCs are known for their heterogeneity, and there are currently no genetic tools available that can achieve both highly efficient and specific gene deletion solely in HFSCs. Given this limitation, *K5-Cre* and *K14-Cre* mouse models remain widely employed in HFSC studies, as these promoters are expressed in basal keratinocytes of both the interfollicular epidermis (IFE) and HF bulge, making them useful for studying HFSC biology. Notably, the *K5-cre* and *K5-Cre^{ER}* gene deletion systems only target the epithelium but not DP of HFs as shown in Figure S1K.

To address whether *Mcl-1* loss specifically affects HFSCs or their niches, we performed several experiments. TransEpidermal Water Loss (TEWL) assays showed no difference between *Mcl-1* knockout and control mice during HF regeneration following depilation (Figure S5A), suggesting that the skin barrier remained intact. Furthermore, *Mcl-1* deletion did not impact the KRT14⁺ basal epithelium or the KRT10⁺ terminally differentiated epidermis, and proliferation within the KRT14⁺ interfollicular epidermis was similar in knockout and control mice (Figures S5B&D). H&E staining and immune marker analysis revealed no signs of inflammation or immune cell infiltration (Figure 4C; Figure S5F), and no significant apoptosis was observed in the IFE (Figure S4A). These findings indicate that *Mcl-1* deletion did not cause defects related to skin barrier disruption or immune response in the IFE.

To specifically assess role of MCL-1 in HFSCs, we used the *K5-Cre^{ER}* system to delete *Mcl-1* after HFSC activation by administering tamoxifen 3 days post-depilation (3 dpd), when HFSCs had already committed to becoming transit-amplifying cells (TACs). Under these conditions, *Mcl-1* was efficiently deleted in the IFE, but HF regeneration remained unaffected in *Mcl-1* knockout mice compared to controls (Figures S4G-K). This suggests that MCL-1 is required only before HFSCs commit to differentiated lineages, providing evidence that hair follicle defects are due to the loss of MCL-1 in HFSCs and not secondary effects in the IFE.

The HFSC compartment is heterogenous, and there is currently a lack of powerful inducible *Cre* systems that can delete genes both efficiently and specifically in all HFSC subpopulations—efficiency and specificity are both crucial. Given the well-recognized heterogeneity of HFSCs in the bulge region, *Krt5- / Krt14-Cre* mouse models remain widely employed to investigate HFSC biology (Hsu *et al.*, *Nat Med*, 2014; PMID 25100530) (Hsu *et al.*, *Cell*, 2011; PMID 21215372) (Kretschmar & Watt, *Cell*, 2012; PMID 22265400). Nevertheless, to explore the possibility of using a more HFSC-specific driver, we generated a conditional knockout model, *Lgr5-Cre^{ER}/Mcl-1^{ff}*, to specifically assess the role of MCL-1 in *Lgr5*⁺ HFSCs. Following *Mcl-1* deletion in the *Lgr5*⁺ cycling HFSC subpopulation (Jaks *et al.*, *Nat Genet*, 2008; PMID 18849992) (Hoeck *et al.*, *Nat Cell Biol*, 2017; PMID 28553937), hair regeneration after depilation remained unaffected (Rebuttal Figure 1). Notably, *Mcl-1* deletion only occurred in a proportion of HF cells in this model. We further crossed *Lgr5-Cre*

mice with *Rosa26-tdTomato* mice to examine the deletion efficiency in different subset of the skin epithelial cells by the *Lgr5-Cre* system. TOMATO expression was found in around 51% and 13% of the HFSCs ($Lin^-/CD49f^+/CD34^+/SCA-1^-$) and the non-defined HF ($Lin^-/CD49f^+/CD34^-/SCA-1^-$) populations respectively, but not in the IFE cells ($Lin^-/CD49f^+/SCA-1^+$). Confocal analysis further confirmed that TOMATO expression was only switched on in a small proportion of HF cells. Collectively, these results suggest either a functional redundancy within distinct HFSC populations or a redundancy of distinct pro-survival Bcl-2 family members for the survival of *Lgr5*⁺ HFSCs and their progeny. We prefer not to include this data in our manuscript as we believe more comprehensive studies with inducible *Cre* system driven by the promoter of different HFSC marker genes are required to precisely map out the HFSC subpopulation(s) that are dependent on MCL-1 for their survival and contribute to the observed HF defects in this study, a process that would require significant time, resources, and budget, and could take a few years to fully resolve.

Rebuttal Figure 1: Loss of *Mcl-1* in the *Lgr5*⁺ HFSC subpopulation does not result in HF regeneration defect

(A) *Lgr5-Cre^{ER}/Mcl-1^{ff}* mice were administered with tamoxifen containing diet from p55 throughout the course of the experiment. A portion of the dorsal skin was shaved and waxed to induce hair regeneration. Skin samples were analysed 15 dpd. (B, C) Comparable hair regeneration between *Lgr5-Cre^{ER}/Mcl-1^{ff}* mice and *Mcl-1^{ff}* as shown by H&E histological sections (B), and confocal imaging (C). Representative confocal images show that a significant proportion of HF cells of *Lgr5-Cre^{ER}/Mcl-1^{ff}* mice maintained the expression of MCL-1 (Left panel), even though *Lgr5-Cre^{ER}* mediated recombination as indicated by hCD4 expression was observed (Right panel). n = 6-7 mice per genotype. (D) Representative confocal images showing that the KRT15⁺ HFSC subpopulation is still present in *Lgr5-Cre/Mcl-1^{ff}* mice. (E, F) Only a proportion of CD34⁺ HFSCs (CD34⁺/SCA-1⁻), and a small proportion of non-defined HF cells (CD34⁺/SCA-1⁻) expresses TOMATO (RFP) in the *Lgr5-Cre^{ER}/tdTomato^{KI/+}* mice, as analysed by FACS (E) and confocal image (F).

4. The apoptotic HFSCs during anagen are quite interesting. How does this phenomenon lead to the loss of the hair follicle structure? Can the authors demonstrate the dynamic of hair follicles that eventually leads to losing the HF structure?

Adult HF are dynamic organs that cycle through three stages: growth (anagen), apoptosis-driven regression (catagen), and relative quiescence (telogen), all of which are regulated by HFSCs (Tumbar *et al.*, *Science*, 2004; PMID 14671312). In the absence of MCL-1, activated HFSCs are unable to survive proliferation stress and are rapidly depleted during anagen. This depletion of HFSCs disrupts the regeneration process, leading to the progressive degeneration of HF structure. The loss of HFSCs has been shown to result in permanent loss of the HF architecture, as the stem cells are essential for regenerating the follicle and maintaining its structure throughout the hair cycle. To demonstrate this dynamic, our data reveal that in *Mcl-1* knockout mice, the rapid depletion of HFSCs during anagen leads to incomplete or failed HF regeneration, ultimately resulting in the loss of the follicle structure. As the follicle lacks the essential stem cell population, it is unable to undergo subsequent hair cycles, resulting in permanent structural loss.

5. Does the *K5-Cre* also target the epidermis? If yes, do they also suffer a loss of epidermal stem cells since the epidermis is under constant self-renewal driven by activated epidermal stem cells?

K5-Cre targets both HFSCs and IFE, where K5 and K14 serve as markers of the basal layer within both the HF and IFE compartments (Blanpain & Fuchs, 2006). In our study, KRT14 remains expressed in the basal layer of the IFE in both constitutive and inducible *Mcl-1* knockout mice, indicating that while *K5-Cre* drives *Mcl-1* deletion in both HF and IFE, the primary impact is on the HF. Additionally, LRIG1, which may also contribute to the IFE, was still present in *Mcl-1* deficient hair follicles post-depilation (9 dpd; Figure S4F).

To further assess the impact of *Mcl-1* depletion on the IFE, we performed TransEpidermal Water Loss (TEWL) analysis using the *K5-Cre^{ER}* inducible *Mcl-1* knockout model. Our results showed no differences in skin barrier

function, structure, or proliferation between *Mcl-1* knockout and control mice. No inflammation, immune infiltration, or apoptosis was detected (Figure S4A & S5F). Upon activation, HFSCs lose basal markers like KRT5 and differentiate into transit-amplifying cells (Hsu & Fuchs, *Cell*, 2014; PMID 24813615). We induced *Mcl-1* deletion after HFSC activation, and HF regeneration remained unaffected, indicating that *Mcl-1* deletion impairs regeneration only if HFSCs are targeted before commitment and differentiation. The lack of effects on the IFE suggests the observed phenotype is specific to the HF and HFSCs.

6. Figure 1C KRT14 should be green in color.

We have now rectified this mistake in the revised manuscript.

Reviewer #1 (Remarks to the Author)

Chin et al. have addressed a number of concerns and made meaningful improvements during the review process. In particular, the addition of the *Mcl-1/Trp53* double knockout model in Figure 6 showing *Trp53* loss can functionally rescue hair defects resulting from *Mcl-1* deletion strengthens this manuscript. If possible, these results should be integrated together with *Bak* rescue experiments to provide a more wholistic view of intrinsic apoptosis (and its suppression via *MCL-1*) during stem cell-mediated hair growth. Efforts to address other major concerns (e.g. hair phenotypes are secondary to barrier defects, more targeted scRNA-sequencing analyses, and ERBB-targeting small molecule specificities) are satisfactory.

We sincerely thank the reviewer for the positive feedback on our manuscript and for acknowledging the meaningful improvements made during the previous revision process, including the recognition of the data on the *Mcl-1/Trp53* double knockout model and how it strengthens the manuscript. We appreciate the reviewer's thoughtful suggestion to integrate the results from the *Mcl-1/Trp53* double knockout model with the *Bak* rescue experiments to provide a more holistic view of intrinsic apoptosis during stem cell-mediated hair growth. The datasets for the two double knockout models are extensive and complex, which makes it very difficult to present together in a single figure. As such, we have rearranged the order of the figures and put them next to each other to improve the cohesion of the two datasets (Figure 6 & 7 in the new manuscript). Moreover, we have also revised the Discussion section to connect and integrate the insights gained from the two models, providing a detailed and unified narrative of intrinsic apoptosis during stem cell-mediated hair regeneration.

This said, there is still a lingering concern about the absence of a phenotype using the *Lgr5-CreER* model to delete *Mcl-1*. While it is possible that low recombination efficiencies are at fault, only two days of tamoxifen prior to waxing may be insufficient to achieve maximal efficiency. Additionally, if crossed with R26-TdTomato, do the authors observe apoptosis specifically in lineage-traced cells? Or their selective depletion over time after initiating hair cycling? If possible, either or both experiments should be performed and importantly, all data with this model included in the manuscript.

We thank the reviewer for raising this important point regarding the absence of a phenotype in the *Lgr5-Cre^{ER}/Mcl-1* model and for suggesting additional experiments to clarify this observation. The *Lgr5-T2A-Cre^{ER}* system (Leushacke *et al.*, *Nature Cell Biology*, PMID: 28581476) used in this study is recommended by Prof. Nick Barker for efficient deletion among the different *Lgr5-Cre^{ER}* stains that have been generated and published by his group. We appreciate the suggestion to use the R26-tdTomato reporter to trace the survival and cell fate of *Lgr5⁺* HFSCs. In fact, we initiated the breeding for generation of the *Lgr5-Cre^{ER}/Mcl-1* model by crossing *Lgr5-Cre^{ER}/tdTomato* mice obtained from Nick Barker with our *Mcl-1^{f/l}* mice, and we are pleased to include the data on the analysis of the *Lgr5-Cre^{ER}/Mcl-1/tdTomato* model as a supplementary figure (Figure S6) in the revised manuscript. Of note, Nick Barker has now been included as co-author in the revised manuscript.

For the *Krt5-Cre^{ER}* model used in this manuscript, mice were treated with tamoxifen from 2 days prior to 5 days post depilation. To achieve maximum deletion of *Mcl-1* in the *Lgr5-Cre^{ER}* model, mice were placed on TAM-

containing diet from -2 days to 15 days post depilation (Figure S6A). Under this experimental condition, approximately 30% of CD34⁺ cells and ~5% of non-CD34⁺ HF cells expressed tdTomato and hCD4 (a surrogate of *Mcl-1* deletion), with no recombination in SCA-1⁺ IFE cells (Figure S6B). Histological analysis revealed no differences in hair regeneration between *Lgr5-Cre^{ER}/Mcl-1^{ff}* and *Mcl-1^{ff}* controls (Figure S6C). Confocal imaging analysis indicates that RFP⁺ cells are mainly localized at the lower bulge and proliferative ORS cells, but not at the matrix which contains transit-amplifying cells. Additionally, deletion of both *Mcl-1* alleles did not affect the distribution patterns and percentage of RFP⁺ cells in HFs (Figure S6D). Cleaved caspase-3 staining was not detected even when *Mcl-1* was deleted in *Lgr5*-expressing cells (Figure S6D). Consistent with the FACS data, only a subset of CD34⁺ HFSCs expressed RFP, suggesting that *Mcl-1* deletion occurred in a limited fraction of this population (Figure S6E). The deletion of *Mcl-1* in the cells of bulge and ORS was further confirmed by MCL-1 and hCD4 staining (Figure S6F and S6G). Collectively, these results indicate that MCL-1 is dispensable for the function and cell fate of *Lgr5*⁺ HFSCs during adult hair regeneration induced by depilation.

As shown in Figure below, *Lgr5* is known to only mark a small subset of cells in the heterogeneous HFSC compartment. A recent paper (Hoeck *et al.*, *Nature Cell Biology*, 2017; PMID 28553937) demonstrates that *Lgr5*⁺ HFSCs are not positioned at the apex of the HFSC hierarchy and more quiescent CD34⁺*Lgr5*⁻ HFSCs can give rise to *Lgr5*⁺ cells and compensate their loss during adult hair regeneration. Further investigations are needed in the future to define the specific HFSC population(s) reliant on MCL-1 for their survival and function in fuelling HF regeneration, which would require significant resources and budget, and could take a few years to fully resolve.

[figure redacted]

Figure: Heterogeneity of the HFSC compartment. Adapted from Schepeler *et al.*, *Development*, 2014; PMID 24961797.

On more minor notes:

The authors should re-word lines 342-344 (“Confocal microscopy further showed that p-ERBB2 expression in hair follicles was high during anagen, with a pattern overlapping MCL-1, while it decreased in catagen and was low in telogen as previously reported, paralleling MCL-1 dynamics (Figure S9A).”). It is misleading as the ERBB2 expression referenced is not in the K14⁺ outer root sheath or HFSC compartment, where MCL-1 is in the anagen hair follicle. As it is, this data is not actually supportive of the hypothesis that p-ERBB2 is upstream of MCL-1 expression, and suggests that one of the other ERBB-family members may be key (eg EGFR).

We thank the reviewer for the observation of the inaccurate description on the section and for the excellent suggestion to clarify the interpretation of the data. We agree that the statement about p-ERBB2 expression overlapping with MCL-1 in the anagen hair follicle may be misleading as it does not specifically reference the K14⁺ outer root sheath or HFSC compartments where MCL-1 is predominantly localized. To address this concern, we have revised the related text to reflect the spatial dynamics of p-ERBB2 expression more accurately

and the potential regulation of MCL-1 levels by other ERBB-family members during different stages of the HF cycle. Please refer to Line405 to Line418 on Page17 and Line549 on Page23.

In line 486 of the discussion, there is a typo: ““mutant strains of with EGFR signaling””.

Thank you for pointing this out. We have corrected the typo in the revised manuscript.

Reviewer #2

The current revised article by the group of Dr. Naiyang Fu and collaborators uncovers an essential role of the pro-survival protein MCL-1 in regulating HF development and adult hair regeneration. PI and collaborators utilize mouse genetics approach to demonstrate that MCL-1 is critical for the survival and function of activated HF stem cells (HFSCs) during hair regeneration. They demonstrate that constitutive deletion of Mcl-1 resulted in gradual hair loss and the eventual elimination of HFSCs.

Additionally, utilizing a Kinase screen, and two tyrosine kinase inhibitor of ERBB signaling (afatinib and erlotinib), that authors found that ERBB pathway acts as an upstream regulator and plays a central function in promoting the survival of activated HFSCs and adult hair regeneration through promoting MCL-1 protein expression (Translational control and not transcriptional).

The results are noteworthy, and the methods are sound. The presented work will be of significance in the field of skin and stem cell biology, HF morphogenesis and cycling.

The authors have successfully addressed all the concerns from the earlier submission. No additional concerns remain.

The results are noteworthy and the presented work will be of significance to the field of stem cell biology and related fields of HF biogenesis and regeneration.

We appreciate the reviewer's thorough and concise summary of the important findings in our manuscript. We thank the reviewer for the insightful comments for us to improve our manuscript in the previous revision. We are pleased to know the reviewer think that we have successfully addressed all the concerns and no more concerns remain.

Reviewer #3

The authors have addressed all my questions, and I have no further inquiries regarding the manuscript. I recommend its acceptance.

Thank the reviewer for recommendation of our paper to be published in *Nature Communications*.